# Decisions are expedited through multiple neural adjustments spanning the sensorimotor hierarchy

Natalie A. Steinemann[1,2], Redmond G. O'Connell[3] & Simon P. Kelly[1,4]

When decisions are made under speed pressure, "urgency" signals elevate neural activity toward action-triggering thresholds independent of the sensory evidence, thus incurring a cost to choice accuracy. While urgency signals have been observed in brain circuits involved in preparing actions, their influence at other levels of the sensorimotor pathway remains unknown. We used a novel contrast-comparison paradigm to simultaneously trace the dynamics of sensory evidence encoding, evidence accumulation, motor preparation, and muscle activation in humans. Results indicate speed pressure impacts multiple sensorimotor levels but in crucially distinct ways. Evidence-independent urgency was applied to cortical action-preparation signals and downstream muscle activation, but not directly to upstream levels. Instead, differential sensory evidence encoding was enhanced in a way that partially countered the negative impact of motor-level urgency on accuracy, and these opposing sensory-boost and motor-urgency effects had knock-on effects on the buildup and pre-response amplitude of a motor-independent representation of cumulative evidence.

[1] Department of Biomedical Engineering, The City College of The City University of New York, New York, NY 10031, USA. [2] Zuckerman Mind Brain Behavior Institute, Columbia University, 3227 Broadway, New York, NY 10027, USA. [3] Trinity College Institute of Neuroscience and School of Psychology, Trinity College Dublin, Dublin 2, Ireland. [4] School of Electrical and Electronic Engineering, University College Dublin, Dublin 4, Ireland. Correspondence and requests for materials should be addressed to N.A.S. (email: ns3058@columbia.edu) or to S.P.K. (email: simon.kelly@ucd.ie)

When situations call for it, animals can prioritize speed over accuracy in their sensory-guided actions. Prominent computational models suggest that sensorimotor decisions are made by integrating noisy evidence representations up to an action-triggering threshold or bound[1,2]. In this framework, speed can be emphasized at the expense of accuracy by lowering this bound[3,4]. Motor-selective neural circuits have been found to implement such adjustments in the form of evidence-independent "urgency" signal components, which non-selectively elevate activity towards unchanged action thresholds. A static component of urgency has been widely observed in raised activity towards both response alternatives before evidence presentation[5–8]. Recent work has revealed that urgency can also have a dynamic component that grows over the course of a decision, decreasing the amount of accumulated evidence required to trigger a response (equivalent to a "collapsing bound" in computational models[9]), which decision makers are also capable of adjusting[6,10–12].

Theoretical work has suggested that urgency is implemented through gain modulation[13–15] and recent work has specifically implicated diffusely-projecting neuromodulatory systems[11,16], suggesting that it may act globally, i.e., at all processing levels of the sensorimotor hierarchy. Thus far, however, adjustments to speed pressure have been almost exclusively examined within neural circuits involved in preparing actions. The extent and nature of adjustments made at upstream and/or downstream processing levels has yet to be determined.

Whether and how speed pressure impacts on the level of sensory encoding remains unclear. While computational modeling[17,18] and neuroimaging[7,19] studies have suggested that sensory representations are either weakened or unaffected by speed pressure, one single-unit recording study reported enhanced evidence representations under speed pressure in the context of visual search[20].

Recent studies have highlighted the existence of abstract, motor-independent processes that intermediate between sensory evidence encoding and motor preparation, and afford flexibility in the mapping of one to the other[21–23]. The impact of speed pressure on such processes is unknown, yet it stands to be highly illuminating on the nature and utility of effector-general decision signals. On one hand, if urgency is applied globally then it should manifest at this abstract stage of integration, hastening the process of cognitive deliberation and not just the preparation of actions. Alternatively, urgency signals may first confluence with evidence at downstream motor levels, thus allowing an unadulterated representation of cumulative evidence to be retained at the motor-independent level.

Here, we employed a novel contrast-comparison paradigm that enables neural dynamics at multiple levels of the sensorimotor hierarchy to be traced simultaneously in humans making decisions under varying response time constraints. Using scalp electroencephalography (EEG), we traced sensory evidence encoding via steady-state visual evoked potentials (SSVEP) reflecting contrast-dependent responses in early visual cortex[24]. We traced motor preparation in effector-selective spectral amplitude changes in the Mu/Beta band (8–30 Hz) over motor cortex[25]. Like decision signals identified in sensorimotor neurons in monkeys and rodents[1,26], Mu/Beta activity exhibits key characteristics of bounded accumulation, namely a build-up rate that scales with the strength of sensory evidence (e.g., dot motion coherence) and a threshold-crossing relationship to response execution[25,27]. Also like sensorimotor decision signals in monkey[6], Mu/Beta activity has been shown to exhibit both static and dynamic components of evidence-independent urgency[11]. To additionally examine effects of speed pressure at the peripheral level of muscle activation, we recorded electromyographic (EMG) signals from both alternative response effectors (left/right thumbs).

To measure upstream, motor-independent representations of cumulative evidence, we traced a recently characterized centro-parietal positivity (CPP) in the event-related potential (ERP). Like motor preparation signals, the CPP has a build-up rate that scales with evidence strength and peaks around the time of the response, regardless of the sensory modality or feature being evaluated[27,28]. However, unlike motor-selective signals, the CPP traces evidence accumulation even when the decision entails no overt action[27] or when the stimulus-response mapping is withheld until after commitment[29]. Finally, the CPP build-up temporally precedes that of evidence-selective motor preparation[28], which together with the above features suggests that it reflects an intermediate, motor-independent level in the decision hierarchy, which receives sensory evidence as input and in turn feeds into motor preparation[30].

Our findings indicate that speed pressure impacts processing at multiple hierarchical levels, but not in a way that is consistent with a global application of evidence-independent urgency. Our results indicate that urgency is applied to both response alternatives at the level of motor preparation and reflected in downstream muscle activation, but is not directly applied at upstream levels. Instead, the neural representation of sensory evidence was enhanced under speed pressure, rendering the alternatives more distinguishable, and this was positively linked to choice accuracy. The knock-on impact of this sensory enhancement was exhibited in a steepening of evidence accumulation reflected in the CPP. Meanwhile, evidence-independent motor-level urgency, which was elevated under speed pressure but appeared to grow at a similar rate under the Speed and Accuracy regimes, limited the amplitude that the CPP reached by the time a response was triggered.

## Results

**Contrast discrimination task**. Sixteen human participants performed a two-alternative forced-choice contrast discrimination task at two interleaved evidence strengths. Subjects reported whether the left- or right-tilted lines in a compound overlay-pattern had a greater contrast by pressing a button with the thumb of the corresponding hand (Fig. 1a). This paradigm allows the isolation of sensory signals that specifically encode the input to the decision process i.e., stimulus contrast, through the contrast-dependent SSVEP evoked by phase-reversing the left- and right-tilted patterns at two different frequencies. On a trial-by-trial basis, participants were incentivized to emphasize decision speed or accuracy, as indicated by a color cue at the beginning of the trial (see Methods and Supplementary Fig. 1). Randomized trial-by-trial cueing was used so that the short-term establishment of pre-decision states in preparation for speed pressure could be examined with respect to a recent, pre-cue baseline.

**Response accuracy decreases with speed pressure and with RT**. Responses were faster and more accurate for greater contrast differences (ANOVA; Reaction Time (RT): $F(1,15) = 90.2$, $p = 9.8 \times 10^{-8}$; accuracy: $F(1,15) = 106.8$, $p = 3.2 \times 10^{-8}$ Supplementary Table 4a-b). Subjects made faster responses ($F(1,15) = 47.8$, $p = 5.0 \times 10^{-6}$) at the expense of response accuracy ($F(1,15) = 23.2$, $p = 0.0002$) in the Speed Regime compared to the Accuracy Regime (Fig. 1b). We next analyzed choice accuracy (proportion of correct trials) as a function of RT ("conditional accuracy function") in order to test for dynamic behavioral signatures of urgency. Extremely fast responses in the Speed regime were especially inaccurate, and beyond approximately 600 ms, accuracy decreased monotonically with increasing RT at a similar rate in both regimes (Fig. 1c). This accuracy decline is consistent

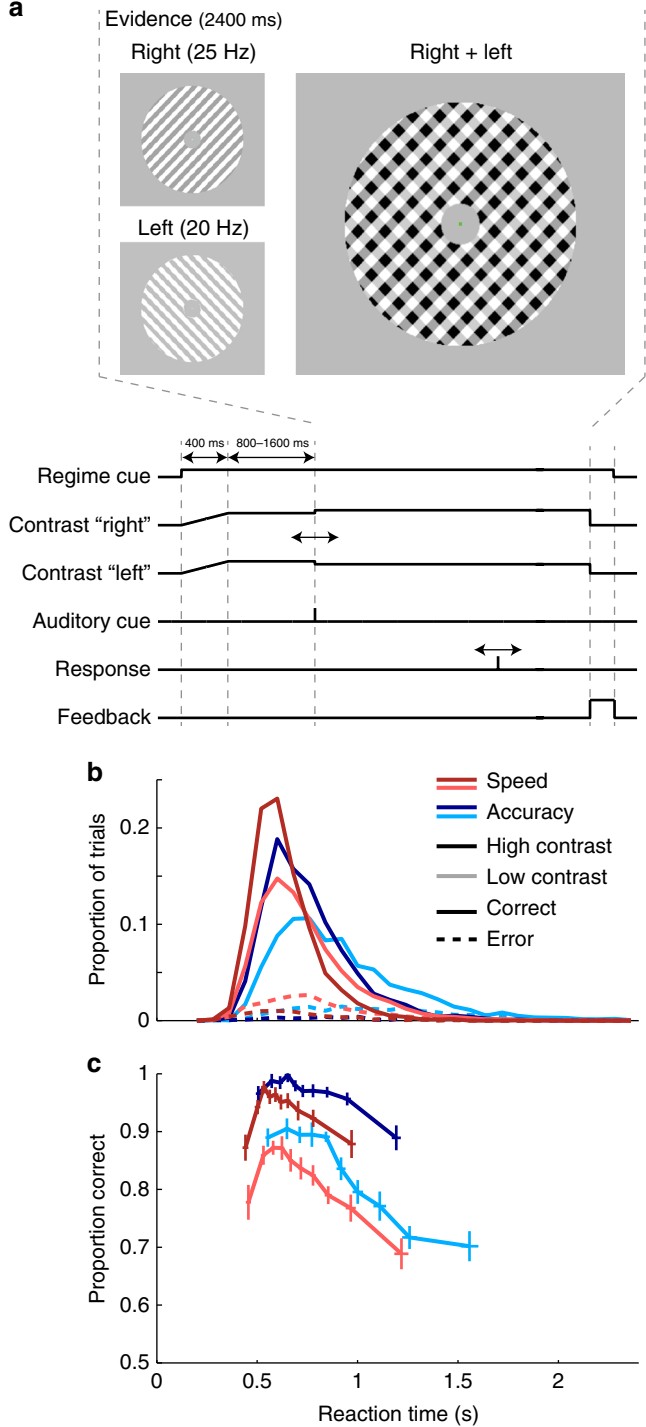

**Fig. 1** Task and Behavior—RT histograms and conditional accuracy functions. **a** Task structure and trial timing. Two overlaid grating patterns tilted 45 degrees to the left and right were phase-reversed at 20 Hz and 25 Hz, respectively. Both gratings initially had an equal contrast of 50% (Baseline), and after a variable delay one stepped to 56% (low contrast) or 62% (high contrast), while the other decreased by the same amount to 44% or 38%. In the example stimulus in a, the right-tilted pattern increased in contrast. This contrast-difference "evidence" was displayed for 2.4 s and evidence onset was marked by an auditory cue to avoid temporal ambiguity. **b** Reaction time distribution for correct (solid) and incorrect (dashed) response trials of high (darker lines) and low (lighter lines) Contrast differences in the Speed (red) and Accuracy (blue) regime. **c** Response accuracy computed as the proportion of correct responses in each of ten reaction time bins separately for each condition. Horizontal and vertical error bars denote the standard error of the mean RT and response accuracy across the 16 subjects. Apart from a low response accuracy for very fast responses, conditional accuracy functions of all conditions declined over RT

of the two phase-reversing gratings (mean difference between left-tilted and right-tilted; Fig. 2b). As expected, the differential SSVEP underwent a change in opposing directions for trials in which the left- versus right-tilted grating was higher in contrast (main effect of "Target Type" (left vs. right) in 3-Way repeated-measures ANOVA, 250–450 ms post evidence onset; $F(1,15) = 26.8$, $p = 0.0028$, Supplementary Table 4d), and this directional amplitude change was strongly modulated by the difference in stimulus Contrast (Contrast (high vs. low) × Target Type interaction: $F(1,15) = 41.2$, $p = 0.00002$). More surprisingly, differential evidence was also boosted under speed pressure (Regime (Speed vs. Accuracy) × Target Type interaction: $F(1,15) = 5.5$, $p = 0.034$, Figure 2b, c). This modulation was significant for all individual 400-ms windows centered from 350 to 550 ms (Fig. 2b, top) and appeared to be limited to the period of decision formation, in that the Regime × Target Type interaction was significant just before the response (response-locked window centered at −50 ms, ANOVA $F(1,15) = 5.7$; $p = 0.031$; Supplementary Table 4e) but not just after (+50 ms $F(1,15) = 2.9$; $p = 0.11$; Supplementary Table 4f; Fig. 2d). SSVEP amplitudes for individual phase-reversal frequencies showed no main effect of speed pressure either before evidence onset (t-tests on window −400 to 0 ms; 20 Hz: $t(15) = 0.4$, $p = 0.68$; $BF_{01} = 3.610$; 25 Hz: $t(15) = 0.5$, $p = 0.61$; $BF_{01} = 3.467$) or in the decision-formation time range over which the above sensory boost effect (Regime × Target Type interaction) was significant (ANOVA, 350–550 ms; 20 Hz: $F(1,15) = 0.6$, $p = 0.46$; $BF_{01} = 4.969$; 25 Hz: $F(1,15) = 0.3$, $p = 0.61$; $BF_{01} = 5.493$; Supplementary Table 4g-h, see Supplementary Fig. 2 for time-resolved analysis). This latter observation indicates that the differential evidence boost did not arise from a general increase in amplitude to both sensory components, but, like the task itself, was truly differential in nature. Further, correct trials were associated with greater differential SSVEP amplitude (increased-contrast minus decreased-contrast grating, 350–550 ms after evidence onset) than error trials (ANOVA; Interaction Correct/Wrong × Target Type: $F(1,15) = 7.7$, $p = 0.014$; Supplementary Table 4i), indicating that the impact of boosting the differential evidence signal was to improve accuracy as well as speed.

with the presence of a dynamically growing urgency component operating in both regimes[11]. The rate of accuracy decline over RT did not differ significantly as a function of Regime (slope of a line fit to response accuracy over 8 RT bins lying beyond each subject's point of maximum response accuracy, ANOVA, $F(1,15) = 1.7$, $p = 0.22$, Supplementary Table 4c), although the evidence for equivalence was not strong ($BF_{01} = 2.197$).

**Enhanced differential sensory evidence under speed pressure.** The neural representation of sensory evidence was quantified as the difference in SSVEP amplitude between the flicker frequencies

**Increased pupil size under speed pressure.** Theoretical work has suggested that sensory representations may be enhanced under

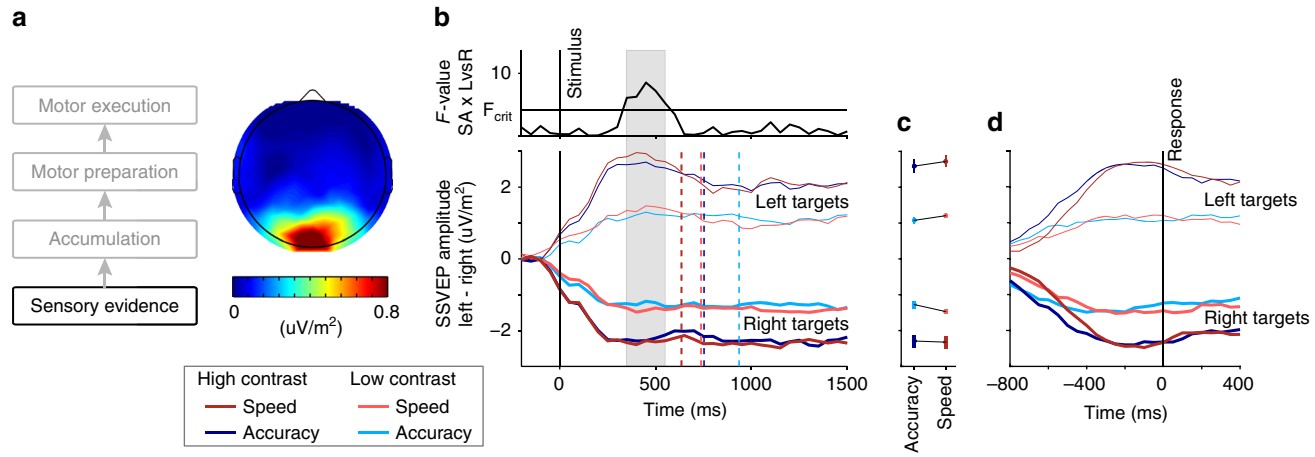

**Fig. 2** Sensory evidence modulations due to speed pressure. **a** Average steady-state visual evoked potential (SSVEP) measured in the 800 ms before stimulus onset was maximal at occipital electrodes around standard site Oz. **b** The difference in amplitude between the steady-state visual response to the left- and right-tilted gratings underwent clear changes in the direction of the Target Type (increased-contrast in the left/right-tilted grating, thin/thick lines), and this directional effect varied in strength as a function of high/low Contrast difference (dark/light shades), and Speed/Accuracy Regime (red/blue). Specifically, there was an increased separation between the stronger sensory representation of the increased-contrast and the weaker representation of the decreased-contrast grating under speed pressure. Upper panel shows the strength (*F*-value) of the Regime × Target Type interaction in a time-resolved fashion, illustrating the time frame during which the differential between left- and right-targets was significantly widened under speed pressure (shaded gray). Dashed vertical lines indicate mean RT of each condition. Note that the gradual initial increase/decrease in the differential SSVEP (−100 to about +200 ms), despite the contrast change stepping instantaneously, is attributable to the fact that SSVEP amplitude at a given time is measured in a 400-ms window centered on that time, and this leads to the apparent deviation even before evidence onset. **c** Mean SSVEP amplitude in the a priori chosen time window between 250 and 450 ms after stimulus onset with error bars depicting the standard error of the mean across the 16 subjects. **d** Differential SSVEP amplitude plotted aligned to the response

speed pressure by increased cognitive effort[31,32]. Motivated by established links between pupil size and cognitive effort and/or arousal[33,34], and the previous suggestions that urgency may be generated by pupil-linked arousal systems[11], we examine effects of pupil size. We found relatively increased pupil size under speed pressure in the 500 ms period prior to evidence onset (*t*-test: $t(15) = 2.3$, $p = 0.040$). This effect increased in magnitude over the course of evidence presentation, as indicated by a significant Regime × Time interaction (ANOVA; 30 50-ms time windows between 0 and 1500 ms after evidence onset; $F(29,435) = 17.6$; $p = 0.0005$; Fig. 3; Supplementary Table 5a). Moreover, greater pupil size (median split within condition; 0–1500 ms after stimulus onset) predicted greater signed SSVEP differences between left and right targets (350–550 ms after stimulus onset; ANOVA: Left/Right × Pupil: $F(1,15) = 10.8$, $p = 0.0050$; Supplementary Table 5b).

**Urgency applied directly at motor level but not upstream.** We traced the dynamics of decision-related buildup in Mu/beta-band indices of motor preparation over the motor cortex of each hemisphere, separately indexing each response alternative (Fig. 4a–d)[27,35], and also in a motor-independent signature of evidence accumulation (CPP; Fig. 4e–h)[27,28]. Both signals exhibited a gradual build-up that scaled with evidence strength (ANOVA; Contrast effect on slope of motor preparation for executed response: $F(1,31) = 9.2$; $p = 0.008$; CPP slope: $F(1,31) = 13.0$; $p = 0.0026$; Fig. 4c, g, Supplementary Table 5c, 5d) and reached their peaks around the time of the response. The build-up rates of both signals also predicted RT independent of evidence strength (CPP slope in 3 RT bins within each condition: $t(15) = -4.7$; $p = 0.0003$; Mu/Beta slope: $t(15) = -6.5$; $p = 1.1 \times 10^{-5}$; Supplementary Fig. 6). However, the two signals exhibited markedly different urgency effects.

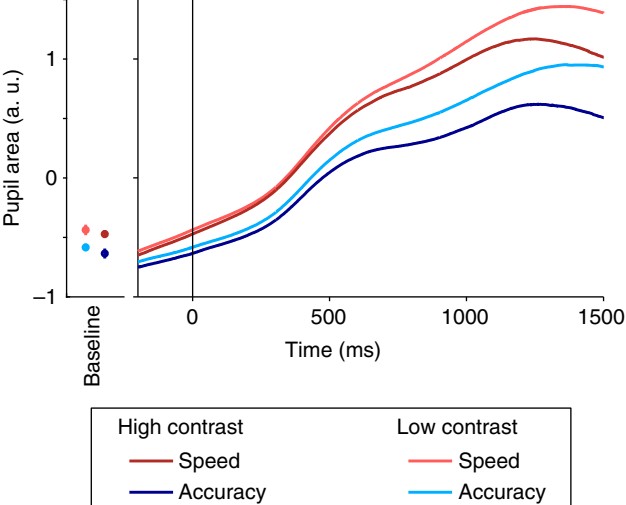

**Fig. 3** Pupil size is modulated by speed pressure. Traces depict mean pupil size plotted over time aligned to evidence onset. Traces are baseline-corrected with respect to the average pupil size in a −500 to 0 ms time window with respect to the onset of the regime-cue. Pupil size was increased under speed pressure (red traces) starting just before evidence onset. The left panel shows the mean pupil size at the onset of evidence presentation for all four conditions with error bars indicating S.E.M. across the 16 subjects. The effect of speed pressure increased in magnitude over the course of evidence presentation, visible in the increasing separation between red and blue traces

Consistent with previous work[5–8,11], motor preparation signals exhibited higher activation (−300 to 0 ms) prior to evidence onset following Speed cues, for both the executed response (linear mixed-effects model (LME): $\chi^2(1) = 11.9$, $p = 0.0006$) and the

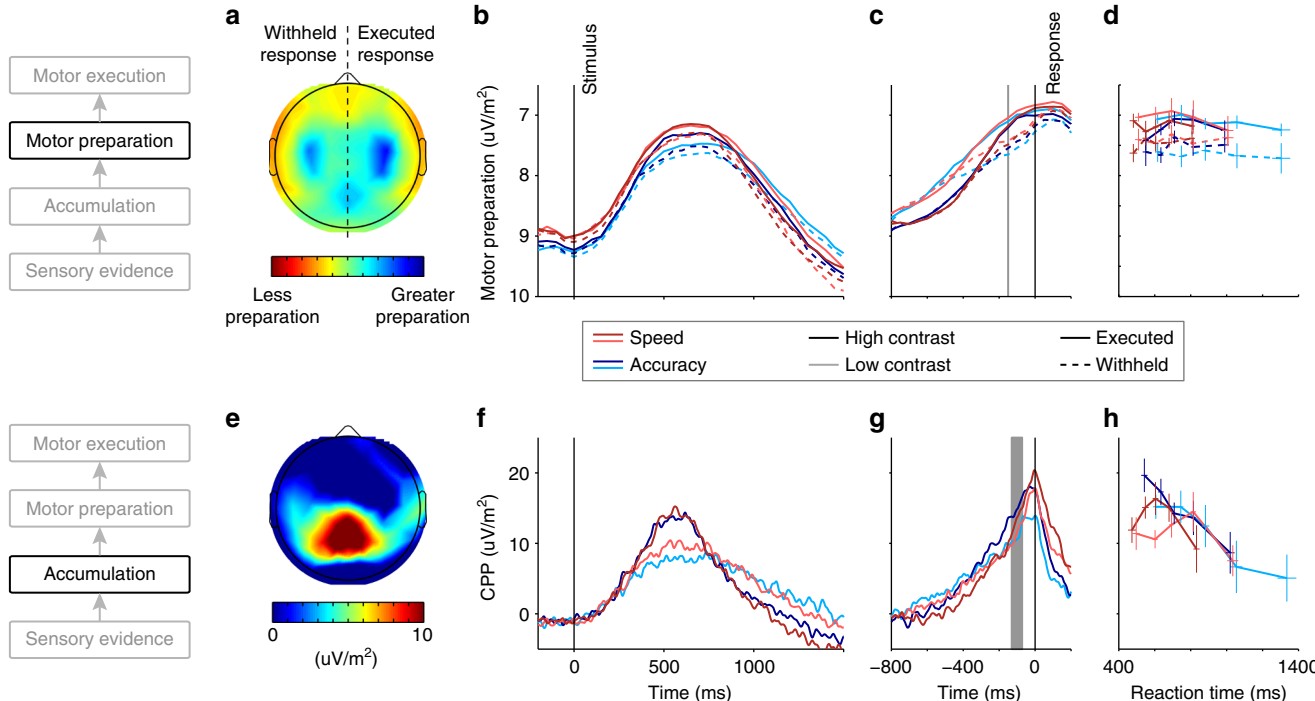

**Fig. 4** Evidence and urgency effects on effector-selective and motor-independent decision signals. **a–d** Motor preparation. **a** Topography of the decrease in Mu/Beta-band amplitude at response, relative to pre-evidence baseline. Topography for left-hand responses is collapsed with a left-right-reversed topography for right-hand responses. **b** Stimulus-locked and **c** response-locked timecourses of Mu/Beta activity reflecting temporally increasing motor preparation (reduced spectral amplitude; note reversed y-axis) for the executed (solid) and the withheld (dashed) response. Spectral amplitudes for each time point are computed in a 300 ms window centered on that time point, resulting in a temporal smoothing effect. The gray vertical line indicates the time point for measurement of pre-response motor preparation (center of 300-ms time window). **d** Pre-response motor preparation at response plotted over RT. The level reached by the motor preparation towards the executed response (solid) is independent of evidence strength (light vs. dark), Speed/Accuracy regime (red vs. blue), and RT. As expected, motor preparation for the withheld response alternative reached significantly lower levels than that of the executed response (main effect of Executed/Withheld response, ANOVA $F(1,15) = 14.3$; $p = 0.0018$ Supplementary Table 5e). Error bars indicate S.E.M. across 16 subjects. **e–h** Motor-independent evidence accumulation. **e** ERP topography around the time of response commitment ($-130$ to $-70$ ms with respect to the button press, gray horizontal bar in **g**), showing a clear centro-parietal positivity. **f** Stimulus-locked and **g** response-locked centro-parietal traces for different evidence levels and Speed/Accuracy regimes, after subtraction of auditory evoked potential (see Methods, Supplementary Fig. 3). **h** CPP amplitude around the time of decision commitment, plotted over RT. Error bars indicate S.E.M. across 16 subjects. Note that y-axis scaling is identical across panels for each of the two decision signals, shown in **b** and **f**, respectively

withheld response ($\chi^2(1) = 16.3$, $p = 5.6 \times 10^{-5}$, Supplementary Table 1a–b), consistent with a common component of static urgency. Further, greater starting levels of motor preparation predicted faster RTs (executed response: $\chi^2(1) = 10.1$, $p = 0.0015$; withheld response: $\chi^2(1) = 3.9$, $p = 0.048$; Supplementary Fig. 5a–b). In the sequential sampling framework, such starting-point variability can explain the fast errors observed in the Speed regime[2] (Fig. 1c). The fact that greater preparation of the withheld response at baseline was also predictive of faster responses indicates a significant degree of correlated variability in motor preparation, consistent with variability in the static urgency component common to both action alternatives. Despite these variations in the baseline, preparation towards the executed response (contralateral Mu/Beta) reached a stable threshold level just prior to response (300-ms window ending at button click) that did not vary significantly as a function of RT, Contrast or Speed/Accuracy regime (Figure 4c, d; LME; all factors $p > 0.05$, all $\mathrm{BF}_{01} > 12$; Supplementary Table 1c), similar to sensorimotor neural activity in monkeys[36,37].

In contrast to motor preparation signals, the pre-evidence amplitude of the CPP, measured relative to a pre-cue baseline, showed no effect of speed pressure (LME: $\chi^2(1) = 1.5$, $p = 0.22$, Supplementary Table 1d), RT $\chi^2(1) = 0.9$, $p = 0.35$ or any other

experimental factor (all $\mathrm{BF}_{01} > 33$). Further, CPP amplitude at the time of decision commitment was not fixed but rather varied across conditions and RT (Fig. 4h). Most strikingly, it decreased over RT in all conditions aside from the very fastest, low-accuracy trials of the Speed regime, mirroring the most prominent feature of the conditional accuracy functions (Fig. 1c) consistent with slower choices being based on a smaller amount of cumulative evidence. Measuring pre-response CPP amplitude in an a-priori chosen 60-ms window centered on the onset of a peri-response motor potential ($-100$ ms; see Supplementary Fig. 4), a linear mixed-effects model indicated a significant decrease with RT ($\chi^2(1) = 12.7$, $p = 0.0004$, Supplementary Table 2a), and a tendency for higher amplitudes in the Accuracy compared to Speed regime that did not reach significance ($\chi^2(1) = 3.1$, $p = 0.078$; Fig. 4h). Subsequent analysis of the EMG data (see below) revealed that this a-priori selected measurement window is in fact delayed with respect to the mean onset of muscle activation ($-135$ ms) and thus likely delayed with respect to the time at which cumulative evidence reaches the value that results in decision commitment and action initiation[28]. We therefore assessed the reliability of these effects by computing the test for all timepoints over an expanded time range, which revealed significant effects of RT and Regime for 60-ms windows centered in the range $-130$ to $+110$

ms and −170 to −130 ms, respectively (Supplementary Fig. 4; see also discussion). These effects did not depend on whether a potential evoked by the auditory cue was removed (Supplementary Fig. 3).

The lack of urgency effects on post-cue CPP amplitude suggests that the motor-independent representation of cumulative evidence is spared from the static urgency influences that are applied at the motor level. The fact that the Mu/Beta signatures of motor preparation for both alternatives launch towards their thresholds upon evidence onset further suggests the addition of a dynamic component of urgency, which can account for the decreasing pre-response amplitude of the CPP for trials with longer RTs (Fig. 4b): If, as the Mu/Beta signals suggest, the ultimate action-triggering threshold is set at the motor level, then an increasing evidence-independent component at that level decreases the amount of evidence that can be accumulated before the motor threshold is crossed. Thus, motor-level dynamic urgency effectively narrows the bounds on the purely sensory-driven representation of cumulative evidence reflected in the CPP.

Based on the observed boost in differential evidence at the sensory level, a steeper build-up in both evidence accumulation and motor preparation would be predicted under Speed pressure. This was indeed observed for within-trial temporal slope, in both the Mu/Beta motor preparation signals for the executed response (measured 350–150 ms prior to response; ANOVA on temporal slope; Speed/Accuracy regime: $F(1,15) = 11.5$; $p = 0.0040$, Supplementary Table 5c) and the CPP (−300 to −50 ms; $F(1,15) = 5.4$; $p = 0.034$, Supplementary Table 5d). Taken together, the pattern of effects on the CPP are consistent with two separate knock-on effects from adjustments made at other levels: the steepened build-up resulting from the boost at the sensory evidence level, and the curtailment of the CPP's pre-response amplitude by motor-level urgency (see Fig. 5).

**Speed pressure effects on muscle activation**. We recorded electromyogram (EMG) signals bilaterally from the thenar eminence while subjects prepared thumb responses to report their decisions. Mean "motor times," quantified as the time from the onset of the response-initiating EMG burst in the response-executing thumb to the button click, were significantly shorter under Speed compared to Accuracy emphasis (Fig. 6a; Speed: −130.2 ms; Accuracy: −139.9 ms; difference: 9.7 ± 6.4 ms; $t(15) = 6.1$, $p = 2.1 \times 10^{-5}$). This was accompanied by a significantly steeper initial rise (LME; $\chi^2(1) = 8.2$, $p = 0.0041$, Supplementary Table 2b) and greater overall amplitude in muscle activation just prior (−100 to 0 ms) to the executed response (Fig. 6b, c; $\chi^2(1) = 18.1$, $p = 2.1 \times 10^{-5}$, Supplementary Table 2c), in line with recent findings that the shortening of response-executing EMG bursts goes hand-in-hand with intensified muscle activation[38].

The parallel signal recordings further allowed us to examine the dynamics of peri-response evidence accumulation within the muscle activation time frame. In both regimes, the CPP peak time relative to button click completion (Fig. 4g; Speed: mean ± SD-6.8 ± 37.9 ms; Accuracy: −44.2 ± 38.4 ms) was much later than EMG onset (Speed: −130.2 ± 16.7 ms; Accuracy: −139.9 ± 17.8 ms), suggesting that even for simple button presses, responses are initiated while the neural correlates of evidence accumulation are still evolving[39,40]. As might be expected given the accelerated muscle activation, the CPP peaked later relative to the click under Speed pressure than under Accuracy emphasis (ANOVA, $F(1,15) = 14.0$; $p = 0.0019$, Supplementary Table 5f). Interestingly, this delayed CPP peak under speed pressure was also observed with respect to EMG

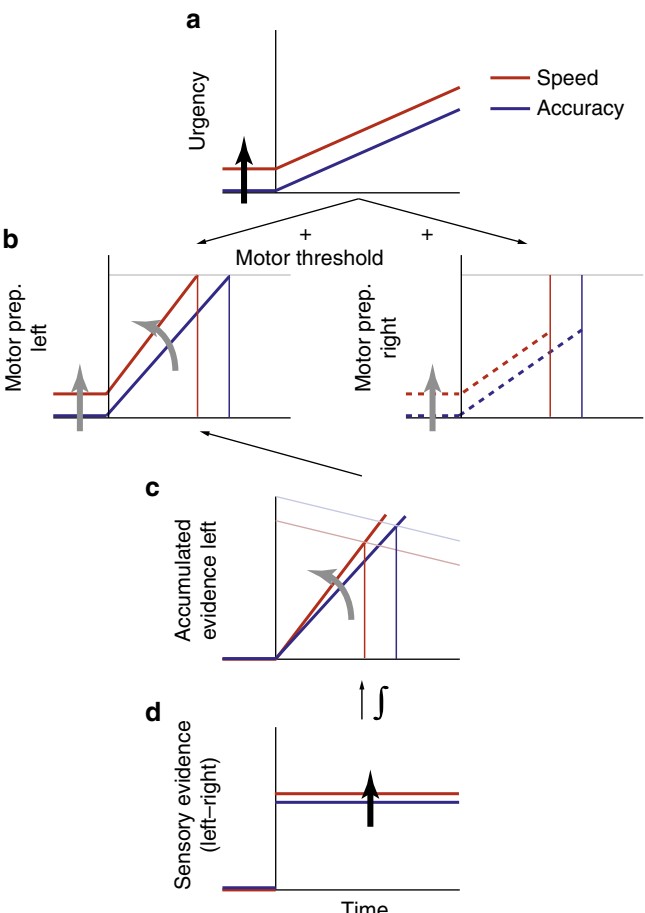

**Fig. 5** Schematic summary of speed pressure effects. The schematic illustrates idealized dynamics at all cortical sensorimotor levels for a typical single trial (left-tilted, correct) in each Regime (Speed in red, Accuracy in blue), representing the primary, active speed pressure adjustments by thick black arrows, and their knock-on impact at other levels by thick gray arrows. In our proposed architecture, neural sensory evidence signals (**d**, reflected by the SSVEP) are temporally integrated at an intermediate, motor-independent level (**c**, reflected by the CPP), which in turn feeds into the motor-level signal representing preparation for the alternative favored by the evidence (**b**, "left" alternative in this example). At the motor level (**b**, reflected by Mu/Beta activity), this favored alternative races against the unfavored alternative (**b**, right), with the first one to cross the motor threshold determining the response choice and timing for that trial. An evidence-independent urgency signal (**a**) additively feeds both motor-level alternatives equally, and grows dynamically over time. Speed pressure has two direct effects: an enhancement of differential evidence represented at the sensory level (**d**; red trace elevated over blue trace), and an upward shift in the urgency component applied at the motor level (**a**; similar rate of increase assumed based on similar conditional accuracy decline, Fig. 1c). These two direct speed-pressure adjustments have knock-on effects that can explain the observed changes in the CPP without necessarily involving any additional adjustment applied directly at that motor-independent level. Specifically, the sensory-level boost steepens the CPP's buildup (**c**, curved gray arrow) so that its post-stimulus amplitude rises higher than for the Accuracy Regime; however, it is the motor-level threshold that ultimately triggers response execution, and the added urgency at that level tends to decrease the level that the CPP can reach by the time of the response. Note that the scales of the axes at different levels are not intended to be consistent, and the size of some effects are exaggerated to aid clarity of illustration

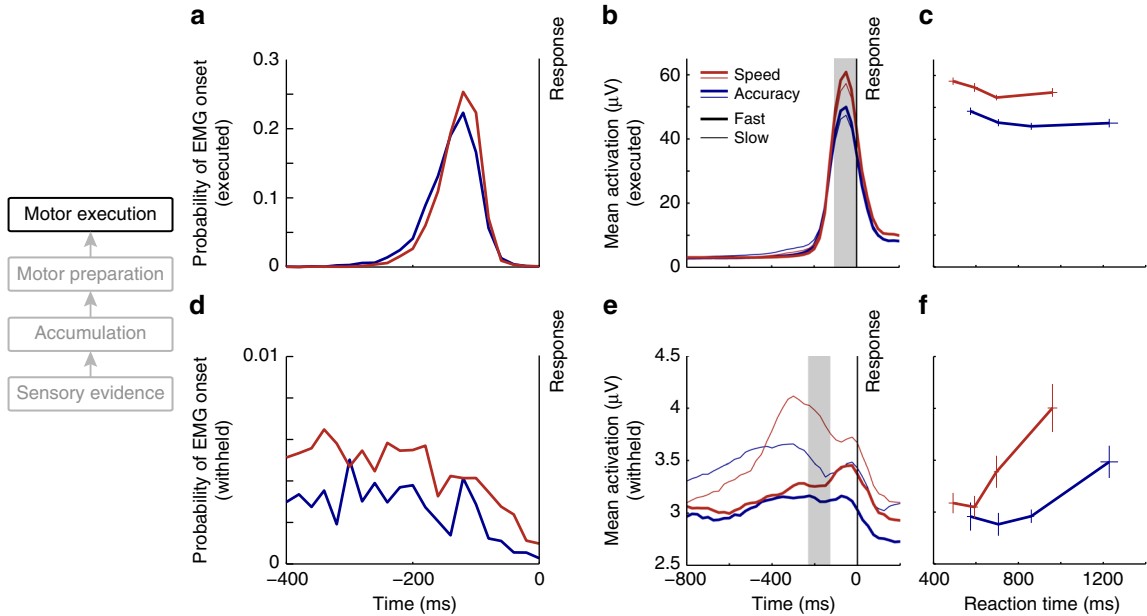

**Fig. 6** Urgency induces graded changes in peripheral muscle activation. **a–c** Electromyographic (EMG) activity in the response-executing thumb. **a** Distributions of "motor times," quantifying the time lag between muscle activation onset and action completion (button click), revealing shorter lags under Speed (red) than Accuracy (blue) emphasis. **b** Muscle activation (mean EMG spectral amplitude over 10–250 Hz measured in 100 ms-time windows) time-locked to the button click response is increased under speed pressure, and for fast (thick traces) compared to slower (thin traces) responses. **c** Muscle activation during response execution (100 ms-time window before click, shaded gray in **b** is increased under speed pressure across all response time bins (Linear mixed-effects model: Regime: $\chi^2(1) = 18.1$, $p = 2.1 \times 10^{-5}$, Supplementary Table 2c), and decreases significantly over reaction time bins (RT: $\chi^2(1) = 5.4$, $p = 0.020$; RT$^2$: $\chi^2(1) = 11.3$, $p = 0.0008$). Error bars indicate S.E.M. across 16 subjects. **d–f** EMG activity in the response-withholding thumb. **d** In the response-withholding thumb, the probability of a muscle activation onset occurring without triggering a motor response (partial burst) is increased under speed pressure across a broad time range. **e** Mean spectral amplitudes measured in two response time bins show that, especially for late responses (thin traces), response-locked traces of muscle activation in the response-withholding hand show increased activation under speed pressure (red traces). **f** With increasing response time, muscle activity in the response-withholding thumb increases both under Speed and Accuracy emphasis in a time window just prior to the mean EMG onset time in the response-executing thumb ($-225$ ms to $-125$ ms, shaded in **e**). Note that activation in the response-withholding hand is plotted on a much smaller scale than that of the response-executing hand. Error bars indicate S.E.M. across 16 subjects

onset (ANOVA: $F(1,15) = 7.8$; $p = 0.014$, Supplementary Table 5g). Although the moment of decision commitment cannot be precisely ascertained, it can be assumed to occur certainly no later than the peak response-executing EMG activation ($-50$ ms), and the delayed peak of the CPP therefore suggests that there is significantly more post-commitment accumulation[41,42] under Speed than Accuracy emphasis. Combined with the steeper buildup under speed pressure, this can explain why the CPP appears to rise to a higher peak following decisions under speed pressure in Fig. 4g.

Consistent with recent reports[38,40], significant muscle activation during decision formation was not confined to the effector ultimately producing the decision report; significant, "partial" (non response-completing) bursts of EMG activity could also be detected in the response-withholding thumb. Such EMG bursts were significantly more prevalent under Speed compared to Accuracy emphasis (Fig. 6d; Speed: $n_{burst} \pm SD = 56.4 \pm 29.5$; Accuracy: $n_{burst} = 42.3 \pm 23.7$; $t(15) = 4.6$, $p = 0.0003$). Further, EMG activation levels of the response-withholding thumb, measured immediately prior to the mean onset time of the response-executing burst to avoid spurious influences from the executing movement on the same mouse ($-225$ to $-125$ ms relative to the button click), were significantly elevated under Speed emphasis (Fig. 6e, f; LME, $\chi^2(1) = 8.1$, $p = 0.0044$, Supplementary Table 2d) and increased over RT (Fig. 6f; $\chi^2(1) = 8.7$, $p = 0.0033$). Since this is the hand to which the evidence is usually opposed, this latter result indicates that an evidence-independent, dynamically growing urgency signal is manifest even at the peripheral stage of motor execution. Following a

similar logic, a recent study found a signature of dynamic urgency reflected in increasing activation of the "losing" hand over RT in cortical motor preparation signals—an effect which accorded with independently conducted behavioral model comparisons[11]. In our data, the excursion (level at RT relative to baseline) of preparation towards the unfavored alternative (ipsilateral Mu/Beta signal, dashed lines in Fig. 4) did not significantly increase over RT (LME, $\chi^2(1) = 2.8$, $p = 0.097$, Supplementary Fig. 5c, Supplementary Table 3a), although evidence supporting equivalence is weak ($BF_{01} = 1.38$; see Discussion).

## Discussion

Our results reveal that speed pressure affects each of the key processing levels necessary for contrast discrimination decisions, from the lowest cortical sensory level to the peripheral level of muscle activation. These modulations across the hierarchy can be accounted for by two principal adjustments that are fundamentally distinct in nature: an evidence-independent urgency contribution applied first at the motor preparation level which creates accuracy costs, and an enhancement of differential evidence at the sensory level that acts to alleviate those costs.

Our finding of boosted differential sensory evidence under speed pressure stands in contrast to classical theoretical assumptions[2,3], recent modeling findings of decreased drift rate consistent with a lower quality evidence representation[18], and fMRI studies finding unchanged[7] or decreased[19] influence of sensory evidence on decision formation under speed pressure and no changes at the sensory level itself[8,19]. Given the transient

nature of the sensory modulations we observed, it is possible that fMRI does not have the requisite temporal resolution to detect such effects, though differing task demands may also play a role. A critical, novel design feature of our paradigm is the choice of the elementary feature of contrast as the discriminandum. This enabled the isolation of separate representations of the two sensory stimuli simultaneously through frequency-tagging and thereby allowed us to distinguish between common, evidence-independent effects characteristic of urgency and differential boosts in sensitivity of the kind that we observed.

Our findings broadly accord with the observation of earlier and stronger spatial selectivity for visual search targets under speed pressure in visual neurons of monkey FEF. However, while visual FEF neurons are modulated by stimulus salience, which represents the evidence for visual search decisions[20], our findings show that speed pressure can also impact low-level sensory representations that form the evidence for simple discriminations requiring no spatial selection. Additionally, whereas speed pressure increased FEF activity somewhat indiscriminately before, during and at the end of decisions in both visual and motor neurons, our SSVEP modulation was strictly evidence-selective (Supplementary Fig. 2). That is, the modulation served to widen the differential activity already driven by the bottom-up stimulus information, but not by turning-up the representation of both alternatives. This effect occurred alongside steepened accumulator signal build-up and in the absence of any apparent background noise modulation reflected in intervening frequencies (Supplementary Fig. 2), and was linked with improved decision performance, all indicating that this modulation lessens the accuracy toll imposed by speed pressure rather than contributing to it. These evidence-selective changes could alternatively be underpinned by increased competitive interactions[43], or narrowing of orientation tuning[44] at the sensory level itself, leading to enhanced contrast sensitivity around the 50% baseline level, or by positive feedback from higher evidence-accumulation and/or motor levels[45].

The differential sensory evidence boost was accompanied by increased pupil size, and there was trial-to-trial covariation between the two. Pupil size has long been linked to generalized factors of effort and arousal[34,46] and central neuromodulatory systems, such as the Locus-Coeruleus Noradrenaline (LC-NA) system[47–49] whose projection sites include sensory areas[50]. These neuromodulatory systems are thought to control global levels of neural gain[34] and dynamic gain modulations during decision formation have been suggested to play a role in optimizing decision making[15,51], and in generating evidence-independent urgency[11] in particular. Our observed evidence-dependent sensory-level SSVEP modulations stood in contrast to the non-selective or evidence-independent sensory modulations predicted by such global urgency influences. This does not preclude that the LC-NA system played a role in our sensory modulations; a growing number of theories of LC-NA function have asserted that its global influences may act to enhance selectivity, since the interaction of locally released glutamate and systemically released NA would act to enhance more active representations while suppressing less active ones[47,52]. Thus, our findings suggest that the impact of pupil-linked neuromodulatory systems on decision making may come in more forms than only accuracy-compromising urgency.

While both systematic and random, behavior-predicting variations were found in baseline amplitudes at the motor level, no such effects were observed at the motor-independent CPP level. Moreover, whereas an invariant action-triggering threshold was observed at the motor level, the pre-response CPP amplitude varied in accordance with urgency influences operating at the motor level (Figs. 4h, 5), qualitatively mirroring the conditional

accuracy functions and consistent with the CPP reflecting an unadulterated representation of cumulative evidence. The quantitative trends in CPP and accuracy were by no means perfectly matched, however, which may relate to the form taken by the underlying evidence accumulation circuits that generate the CPP on the scalp. The CPP manifests as a positive deflection regardless of the decision alternative presented[28], even when the incorrect response is chosen (significant positive amplitudes for errors: $t$ $(15) = 3.63$, $p = 0.0024$; see also[53]) and when falsely reporting the detection of a target[27]. This means that any proportional relationship between mean centro-parietal amplitude at response and response accuracy would break down when the latter approaches chance level. In particular, for longer RT trials in the low contrast condition, which would be characterized by weak evidence coupled with narrowed effective bounds, many errors may be associated with significant diffusion in favor of the incorrect alternative, which would translate to relatively elevated, positive CPP amplitudes at response even though response accuracy is greatly reduced.

An ongoing debate has centered on whether slow errors are, in general, better explained by a collapsing decision bound[54] (equivalent to additive, evidence-independent urgency[6,10,11]) or by drift rate variability[2,4,55]. The collapsing CPP-amplitude effect observed here clearly points to the presence of the former mechanism. This does not preclude that there is some amount of drift rate variability, but renders it an unlikely primary driver behind slow errors. It is also noteworthy that this collapse in CPP amplitude contrasts strikingly with observed patterns in continuous monitoring tasks, where CPP amplitude is stable across RT in a similar way to motor signals[27,28]. In our proposed framework, this would indicate an absence of time-dependent dynamic urgency in these tasks, which is plausible given that subjects are unable to predict the onset of the sensory evidence "targets" and, indeed, are unaware of missing them.

Speed emphasis primarily impacted on starting levels of motor activity in our data, with no strong signs of any change to the rate of growth of dynamic urgency as has been reported previously[6,11]. This is likely attributable to the fact that a response deadline (2.4 s) was imposed even in the Accuracy regime, the fact that difficulty levels were interleaved in both regimes, for which collapsing bounds represent the optimal strategy[9], and/or the interleaving of speed and accuracy trials rather than block-by-block manipulations that typically induce greater adjustments.

We sought additional evidence for the application of a dynamic component of urgency at the motor level by examining motor preparation for the withheld response as a function of RT. Using similar reasoning to another recent human neurophysiology study[11], it could be expected that if Mu/Beta activity contralateral to the withheld response receives the same common urgency input as the signal for the executed response, then it would grow to reach higher levels of preparation at the time of response on trials with longer RT. Although this test indicated a significant effect at the peripheral level of muscle activation, it did not reach significance for cortical Mu/Beta activity. A limitation of this test is that it uses variations in amplitudes across trials with increasing RT as a proxy for dynamics unfolding during a typical single trial, which cannot be counted upon. The relationship between the level reached by the "losing" motor alternative at response and RT would depend critically on which features of the decision process can randomly vary across trials and thereby influence RT, and also on what exactly drives the buildup of motor preparation for the wrong alternative—e.g., urgency alone, negative sensory evidence, automatic co-activation with the opposite limb, or a mixture of these factors—none of which is fully known at present. Future experiments using an expanded set of difficulty levels including a zero-evidence condition may furnish more

straightforward markers of dynamic urgency at the motor level itself, comparable to recent demonstrations related to motion discrimination in monkey neurophysiology[6,10].

Our finding that the CPP represents cumulative evidence unaltered by urgency influences acting at the motor level offers new insights into the brain's architecture for decision computations. Abstract choice representations have been proposed as an efficient means for the brain to flexibly route sensory information to goal-relevant motor regions[21,22,28] and, in fact, the suggestion was made in very early work that such signals may not be influenced directly by speed pressure[56]. Interestingly, the lateral intraparietal area of the monkey has been shown to encode not only signals that are jointly driven by cumulative evidence and urgency similar to motor preparation measures here[6], but also signals of a more abstract, category-selective nature[22]. However, it is not known whether these category-selective signals specifically exhibit evidence accumulation dynamics and future research will be required to determine how such signals relate to the abstract process reflected in the CPP.

The time between the onset of substantial muscle activation and the button click was significantly decreased under speed pressure. At first glance, this appears consistent with a decrease in the additive "non-decision time" component of RT as found in computational model fits in some studies[57]. However, a growing number of studies have demonstrated that action execution, even for simple button clicks, is not deferred until a decision bound is crossed but rather can be dynamically shaped by the ongoing evidence-accumulating decision variable[40]. The existence of partial EMG bursts underlines that EMG onset does not mark complete commitment or a "point of no return." Assuming a fixed mapping of the decision variable to EMG activation, the decreased motor time could arise from either a decreased decision bound or steeper accumulation. The fact that our EMG signals rise more steeply and reach a higher amplitude under Speed pressure points to the latter explanation.

In contrast to our findings of shortened and intensified muscle activation, work in primates suggests no differences in saccade velocity as a function of RT or speed pressure[20], implying that this finding does not necessarily generalize to all actions, although species differences may also be relevant here. While the muscle activations required to initiate saccadic responses stand in direct conflict with one another for different response alternatives, most other response modalities allow for the simultaneous preparation of multiple responses at the muscular level with a much lower degree of antagonism[58]. The presence of significant muscle activation and discrete EMG bursts in the response-withholding effector in our data presents strong evidence that subjects were indeed preparing both responses simultaneously, and to a greater degree under speed pressure.

Taken together, our results serve to highlight the value of the global systems-level view over decision making mechanisms afforded by noninvasive assays in humans employed in the context of purpose-designed paradigms. We uncovered multi-faceted adaptations to speed pressure at the sensory, evidence accumulation, motor preparation, and motor execution levels as well as dynamic neuromodulatory influences reflected in pupillometry. As demonstrated here, multi-level signal tracking affords the ability to test predictions of a hypothesis at multiple hierarchical processing levels, or to adjudicate between alternative interpretations of an effect at one level by testing predictions that apply to other levels. More generally, our findings underscore the emerging imperative to move from one-dimensional decision models to more neurally-based models embracing the hierarchical, interactive, and flexible nature of real neural systems accomplishing adaptive decisions[1,51,59], and highlight that neural recordings in humans can act as a strong guide.

## Methods

**Participants.** Sixteen participants (five male, mean age ± SD = 27.1 ± 4.7 years) gave written and informed consent to partake in this study. All participants had normal or corrected-to-normal vision, and no history of psychiatric illness or head injury. All procedures were approved by the Institutional Review Board of the City College of New York and were in accordance with the Declaration of Helsinki. Subjects gave informed consent and were compensated for their participation with between $9/hour and $15/hour depending on their task performance.

**Contrast discrimination task.** Participants were asked to perform a discrete two-alternative forced-choice contrast discrimination task. Visual stimuli were presented on a gamma-corrected CRT monitor (Dell M782) with a refresh rate of 100 Hz inside a dark, sound attenuated, and radio frequency interference-shielded room. Visual and auditory stimulus presentation was programmed in Matlab (MATLAB 6.1, The MathWorks Inc., Natick, MA, 2000) using the PsychToolbox extension[60]. Participants were seated at a viewing distance of 57 cm from the monitor, and asked to fixate on a central fixation point. The participants' task was to judge which of two overlaid, orthogonal gratings was greater in contrast. The imperative stimulus was an annular pattern with an inner and outer radius of 1° and 6° of visual angle, respectively, presented centrally on a gray background with the same mean luminance as the stimulus (65.2 cd × m$^{-2}$). The pattern consisted of two overlaid square-wave gratings (spatial frequency = 1 cycle per degree) tilted at −45 and +45 degrees relative to vertical, which were phase-reversed at 20 Hz and 25 Hz, respectively (Fig. 1a). To minimize adaptation to the stimulus over the course of a block, the phase of the gratings was shifted by 0.5 cycles per degree on consecutive trials. Flickering the left- and right-tilted patterns at two different frequencies generated contrast-dependent, distinguishable oscillations at the respective frequency over visual cortex, so that the amplitudes of both patterns' neural representations could be measured simultaneously. After 200 ms of presenting just the fixation point (2 × 2 pixels), trials began with a linear fade-in of this stimulus from 0% contrast to 50% contrast of both gratings over 800 ms, in order to eliminate task-irrelevant visual potentials evoked by sudden onsets. When the stimulus had reached 50% contrast, a reward-regime cue was presented in the form of a change in the color of the fixation point (colors randomized across subjects). This regime cue remained on the screen throughout the duration of the trial. During the following baseline stimulus presentation (800, 1200, or 1600 ms in a pseudorandomized order) the contrast of both gratings was held at 50%. The contrast of one grating was then stepped up to 56% (low) or 62% (high) and that of the other grating simultaneously stepped down to 44% (low) or 38% (high), and this contrast differential was held fixed for a full 2400-ms evidence-presentation interval. Evidence onset was marked by a simultaneously onsetting 100 ms-tone (10 kHz, 5 ms fade-in/fade-out) to exclude any temporal ambiguity about evidence onset. Participants indicated that the left-tilted or right-tilted grating was higher in contrast by clicking a mouse button with the thumb of the corresponding hand. At the end of this interval, feedback was provided in the form of the number of points won on the current trial presented close to fixation for 200 ms alongside a tone whose pitch was proportional to this number of points and whose length indicated whether the response given had been correct (100 ms/250 ms beep for correct/incorrect responses, double beep for responses after deadline). After every 10 trials, participants received an extended feedback in the form of an information screen stating the number of points won on the last 10 trials as well as the number of points won in the current experimental block to that point. Within experimental blocks each trial's evidence-onset delay, target-direction, contrast level, and Speed/Accuracy regime (see next paragraph) was assigned pseudorandomly.

**Speed/accuracy regimes.** This contrast discrimination task was performed in four different reward regimes, where particularly fast or accurate responses were rewarded highly on Speed or Accuracy-trials, respectively. Reward conditions included (1) response time (RT)-independent rewards up to a late deadline which coincided with evidence offset (2.4 s, "Accuracy deadline"), (2) RT-independent rewards up to an early deadline at 1 s after evidence-onset ("Speed deadline"), (3) RT-dependent rewards, which decreased at a low rate (−4.2pts/s, "Accuracy slope"), and (4) RT-dependent rewards, which decreased at a high rate (−50pts/s, "Speed slope"). Rewards as a function of response time are displayed in Supplementary Fig. 1a,b. These reward regimes were initially designed to enable exploration of differences in speed-adaptation mechanisms in the decreasing-reward versus deadline regimes. Through extensive piloting the exact temporal deadlines and rates of reward decrease were adjusted to match mean RTs across the two different methods of Speed/Accuracy emphasis manipulation. In each experimental block, the two regimes for a single reward manipulation method were randomly interleaved for comparison (i.e., either "Accuracy deadline" vs. "Speed deadline", or "Accuracy slope" vs. "Speed slope"). Fifteen participants completed 16 blocks of 60 trials, and one subject completed 24 blocks of 40 trials. Subjects were instructed to try to maximize their points won in every experimental block, as their monetary reward after the experiment was calculated as a function of the sum of points won in four randomly chosen blocks.

**Behavioral analysis.** Participants' behavior was evaluated based on RT distributions for correct and incorrect responses. As an initial step, we determined whether

there was a significant difference in RT distributions between the deadline and slope conditions of the two different Speed/Accuracy regimes ("Speed deadline" vs. "Speed slope", and "Accuracy deadline" vs. "Accuracy slope") using Kolmogorov–Smirnov tests. Since these tests revealed no significant differences between RT distributions in any experimental condition, all consecutive analyses were performed on data pooled across the deadline and slope methods, but separately for Speed and Accuracy emphasis. Conditional accuracy functions were computed as the proportion of correct trials within RT deciles. We examined conditional accuracy functions in order to obtain a RT-dependent measure of the amount of sensory evidence accumulated before a response was initiated, and to test for a decline in accuracy consistent with the application of a dynamic (growing) component of urgency[11]. To statistically test for potential differences in the rate of decline of response accuracy over the slower-RT trials for which this decline was evident, as has been found previously[11], we estimated the negative slope of the accuracy decline beyond the point of maximum performance in each condition. Specifically, we first pooled all trials within a subject across Contrast levels and Speed/Accuracy Regimes and computed the percentage of correct responses within six RT bins. We then identified the RT bin with the greatest average response accuracy and used all trials with response times longer than the mean RT of that bin (cut-off RT). We divided those trials into eight RT bins to compute the slope of a line fit to accuracy against RT and finally computed a 3-way repeated-measures ANOVA on these slope values with factors of Speed/Accuracy Regime, Left/Right Target Type, and Contrast (see Statistical Analysis section below for justification of all statistical test choices). To verify that our conclusions did not depend on the manner in which we defined the point of maximum performance, we repeated the above analysis using 5–10 RT bins to determine the cut-off RT and also with the maximum performance point determined within each individual condition as well as across the conditions pooled, and the outcome was always the same in that there was no main effect of Speed/Accuracy Regime on the conditional accuracy function decline (peak performance determined in 5 RT bins: ANOVA, $F(1,15) = 0.21$, $p = 0.65$; 7 RT bins: $F(1,15) = 0.12$, $p = 0.73$; 8 RT bins: $F(1,15) = 2.67$, $p = 0.12$; 9 RT bins: $F(1,15) = 0.15$, $p = 0.70$; 10 RT bins: $F(1,15) = 0.57$, $p = 0.46$; individual conditions: $F(1,15) = 0.59$, $p = 0.45$).

**Data acquisition and pre-processing.** Continuous Electroencephalogram (EEG) and EMG were acquired using a 96-channel actiCAP system and Brain Products DC amplifiers (Brain Products GmbH, München, Germany) at a sample rate of 500 Hz. Ninety-three channels were used for a customized EEG montage including standard site FCz used as the online reference. The remaining four electrodes were used for recording EMG from the thenar eminence of the left and right thumb. Simultaneously, eye gaze and pupil size were acquired continuously using an EyeLink 1000 (SR-Research) eye tracker. Data were analyzed offline using in-house Matlab scripts in conjunction with data reading routines and topographic mapping functions of the EEGLAB toolbox[61].

EEG and EMG data were detrended linearly offline within each experimental block. Potentials in each EEG electrode were further re-referenced to the average of all EEG channels, a Hamming low-pass filter with a cutoff frequency of 45 Hz was applied, and noisy channels were detected based on their elevated signal variance with respect to the rest of the channels, and interpolated for individual blocks using spherical spline interpolation (an average of $0.57 \pm 0.39$ channels per block). Individual trials were rejected from the analysis if the amplitude of a channel of interest exceeded 90 μV or any electrode's potential exceeded 200 μV at any time point before the response. All EEG data were converted to current source density (CSD)[62] to increase spatial resolution and specifically to reduce spatial overlap between the CPP and the fronto-central negativity[28]. ERP were then extracted from the EEG, EMG and pupillometry data for two different epochs: regime cue-locked epochs were extracted from 500 ms before the onset of the regime cue to 4000 ms thereafter, and target epochs spanned the 1000 ms before and the 3200 ms after evidence onset. EEG epochs were baseline-corrected relative to the 100 ms interval preceding the regime-cue, or evidence onset, respectively, and trials were rejected if the delay between the visual contrast change on the screen and the tone marking this evidence onset for the participant exceeded 30 ms, which occurred on less than 2.5% of trials. Due to the longer time scales associated with changes in pupil size, event-related pupil size waveforms were baseline-corrected with reference to the 500 ms prior to the onset of the regime-cue for both the cue-locked and the evidence onset-locked epochs. Response-aligned traces in all modalities were derived by extracting epochs from −1000 ms to 600 ms relative to the response on every trial. Spectral EEG measures were extracted from both stimulus-aligned epochs and response-aligned epochs through short-term Fourier transforms using boxcar windows of 300 ms (for Mu/Beta, 8 to 30 Hz) or 400 ms (for SSVEP, 20 and 25 Hz) measured in steps of 50 ms.

**Statistical analysis.** For statistical tests evaluating the dependence of a variable on a single categorical independent variable, we computed paired t-tests. To test for significant differences in entire RT distributions between Speed and Accuracy regime manipulations ("Slope" vs. "Deadline") we used Komolgorov–Smirnov tests.

For all statistical tests involving multiple independent variables, we computed linear mixed-effects models on single-trial data if RT was a relevant independent variable, and otherwise used repeated-measures analyses of variance (ANOVA) on

subject-averaged data. The use of linear mixed-effects models with the factor of RT was necessary to disentangle effects of RT arising from time-dependent influences within a trial (e.g., dynamic urgency) from other experimental factors that affected RT (e.g., Contrast, Speed/Accuracy Regime). To take a hypothetical example, if an identical collapsing bound were applied in the Speed and Accuracy Regime and it was through some mechanism other than bound adjustment that RTs were longer in the Accuracy Regime, the fact that the bound would have collapsed further with more elapsed time would mean that a basic comparison of an accumulator signal amplitude across Regimes would exhibit a difference, leading to an RT effect that masquerades as an amplitude (bound) effect. Controlling for such factors requires inclusion of the absolute value of RT as an independent variable rather than its rank within condition (e.g., quantile 1, 2, etc.), necessitating the use of linear mixed-effects models.

The conditional accuracy functions (Fig. 1) revealed that response accuracy was particularly low for very early responses, peaked at approximately 600 ms and thereafter steadily decreased with increasing RT. This suggests that two distinct mechanisms may cause fast and late errors, respectively, and thus we allowed for non-monotonic relationships with RT in all linear mixed-effects models by including an $RT^2$ regression term in addition to RT itself. All linear mixed-effects models further included the fixed effect factors of Speed-vs.-Accuracy emphasis (Regime), stimulus Contrast, and Left-vs.-Right Trial Type to maintain consistency across measures. For both RT and all neurophysiological measures the z-score was computed before being entered into the model. For measures of motor preparation this z-score was computed separately for the left and right hemisphere. For the SSVEP and the CPP, the factor Left/Right distinguished trials based on whether the sensory evidence supported Left or Right choices. For motor level variables such as Mu/Beta amplitude and thumb EMG, Left/Right referred to whether subjects responded with their left or right hand.

In the linear mixed-effects models, we included random intercept terms to account for inter-subject variability in the dependent measures as appropriate for a repeated-measures design, and we included random slopes factors, which allow for differences in effect size across subjects, where their inclusion improved the model fit as assessed through systematic model selection. In contrast to a blanket policy of including all random slope terms by default[63], this data-driven approach has been demonstrated to retain the protection against type I errors while also avoiding increased Type-II errors[64,65]. Specifically, we used an iterative process, in which we first computed a linear model without any random slopes and tested whether the model's fit could be significantly improved by including a random slope across each individual fixed effect factor, assessed using chi-squared tests. We provide an exhaustive account of the step-by-step results of this iterative model selection process in the statistical analysis details section of the Supplementary Methods and Supplementary Table 1. Once the final model was established, we tested whether each fixed effects factor had a significant influence on the dependent variable by testing whether the fit of the model significantly worsened if it were excluded, providing the chi-square metrics stated throughout the results. For all tests we used an alpha level of 0.05.

While there were many tests carried out in total, our statistical analyses centered on a relatively small number of primary tests, each designed to verify the central hypothesis that behavioral speed pressure adjustments are mediated by a single global "urgency" influence that acts on neural activity at all processing levels. We then conducted a larger number of follow-up tests designed to verify the reliability of the outcome of each one of these primary tests. For example, the observation of increased SSVEP differential under speed emphasis was followed up and supported by testing for an increase in the rates of evidence accumulation (CPP) and motor preparation (Mu/Beta) as predicted by this effect, as well as a relationship between SSVEP and choice accuracy (see results section Enhanced differential sensory evidence under speed pressure). Thus, the nature of these follow-up tests is such that their multiplicity lessens the likelihood of type I errors, rather than increasing it. It should also be noted that our analyses of the behavioral data and of baseline Mu/Beta-band activity constituted replications of previously well-established effects[6,8,11]. Corrections for multiple comparisons were therefore not applied in order to avoid an unnecessary inflation of Type-II error risk.

In six cases where our statistical analyses indicated no significant differences across experimental conditions (amplitude of the CPP before evidence onset, mean motor preparation for the executed response at the time of response, excursion of motor preparation for the withheld response, and slope of declining accuracy, and the effect of Speed/Accuracy emphasis on single SSVEP frequencies before or during evidence presentation), we computed Bayes Factors ($BF_{01}$) to quantify the relative probability of the null-hypothesis being true and we report the smallest of these values where there were multiple independent variables[66]. Bayes Factor computations were executed in JASP (JASP Team (2018). JASP (Version 0.9)). The Bayes Factor quantifies the ratio of the probability of the data given one hypothesis being true and the probability of the data given another hypothesis being true. When the null-hypothesis is being tested against alternative hypotheses, greater Bayes Factors therefore indicate greater evidence towards the absence of an effect.

**Neural signatures of sensory evidence representation.** The cortical representation of visual evidence was quantified as the difference in SSVEP of the two target frequencies of 20 and 25 Hz averaged together with their respective first harmonics (40 and 50 Hz), and normalized to their respective neighboring frequency bins at standard site "Oz". SSVEPs were measured on a single-trial basis

using a standard short-time Fourier transform with a boxcar window size of 400 ms, fitting an integer number of cycles of both the 20-Hz and 25-Hz components, with a step size of 50 ms. 3-Way repeated-measures Analyses of Variance (ANOVAs) were used to determine the effect of the imposed Speed/Accuracy Regime, stimulus Contrast (high vs. low) and Target Type (left-tilted increase vs. right-tilted increase) on the differential evidence signal (20 Hz minus 25 Hz spectral amplitude). We computed this ANOVA first on the mean SSVEP within an a priori selected time window of 250–450 ms; specifically, on the mean of five 400-ms time windows centered on time points between 250–450 ms post evidence onset. Sphericity was confirmed for all inputs to this and all other ANOVAs using Mauchly's test. For visualization, these SSVEP amplitudes were baseline-corrected to the 400 ms preceding evidence onset. Having established a differential modulation effect in a Regime × Target Type interaction in this time range, we examined the temporal extent of the effect by repeating the same ANOVA on all individual time windows after evidence-onset, and plotting the timecourse of F-values (Fig. 2b, top). To determine whether the observed SSVEP modulation effects were specifically invoked during decision formation related to active stimulus evaluation, we repeated the same ANOVA on a response-locked time window just prior (−50 ms) and just after the button click (+50 ms).

For the subsequent analyses examining the relationship between the SSVEP and both behavior and pupil diameter as well as amplitude changes of individual target frequencies (20 and 25 Hz), we defined our SSVEP dependent measure as the mean amplitude in the time range for which the Regime × Target Type interaction was significant, 350–550 ms. To assess whether speed pressure enhanced both stimulus frequencies even before evidence onset, as may be predicted by an urgency influence similar to that acting at the motor level, we measured the spectrum in the 400-ms time window preceding evidence onset and computed t-tests on each individual frequency. To further test whether such a global enhancement common to both alternatives was exerted during evidence processing, we computed the spectral amplitude of the individual phase-reversal frequencies (20 and 25 Hz) as well as an intermediate frequency (22.5 Hz) in 400-ms time windows spaced at 50 ms starting from −200 ms. For each of these time windows, we computed a 3-Way ANOVA with the factors Speed/Accuracy Regime, stimulus Contrast (high vs. low) and Target Type (left-tilted increase vs. right-tilted increase). While the effect of Contrast on the differential SSVEP amplitude was measured as an interaction with Target Type, putative urgency influences common to both frequencies can be measured as a main effect of Regime (Supplementary Fig. 2).

**Neural signatures of evidence accumulation.** A CPP previously linked to evidence accumulation[27] was measured at standard site "Pz". Because the onset of evidence was marked by a readily-audible tone, which itself generates a stereotyped auditory evoked potential, we employed a residual iteration decomposition (RIDE) algorithm to discriminate any strictly stimulus-locked auditory component from response-locked, decision-related signals[67]. The algorithm exploits the variability in response latency across single trials to iteratively fit a stimulus-locked and a response-locked component to the data. For each individual subject, the algorithm generates patterns of activity to represent a stimulus-locked sensory-evoked potential and a component temporally locked to each individual trial's response time. On each consecutive iteration, the two components are then modified based on discrepancies between the superposition of the generated components and the data until no further improvement in the fit can be achieved. Critically, the algorithm is naive to all stimulus conditions, and the resulting stimulus-locked component is constrained to be invariant across all trials within each individual subject, so that differences in potentials evoked by stimulus contrast or Speed/Accuracy regime were left untouched by this method. This stimulus-locked auditory evoked potential was then subtracted from the evoked potentials of all trials.

From the resulting CPP we obtained three single-trial measures: baseline activity before evidence onset, buildup rate, and peri-response amplitude. The level of baseline activity was quantified as the average potential in the 50 ms before evidence onset, with epochs baseline-corrected with respect to the 100 ms preceding the regime cue, and statistically assessed via a linear mixed-effects model (see statistical analysis section above). Neural measures and RTs were always z-scored within subjects before being entered into the models. The rate of rise of the CPP was measured as the slope of the response-locked traces between −300 and −50 ms with respect to the response, chosen to capture the period of evidence accumulation on the vast majority of trials. The impact of stimulus Contrast and Speed/Accuracy regime on this rate of rise was established through a 2-Way repeated-measures ANOVA. The CPP amplitude at response was measured in a 60-ms window centered on the inflection point of the lateralized motor potential traced over contralateral motor cortex (−130 to −70 ms, Supplementary Fig. 4), and statistically assessed via a linear mixed-effects model. This time window for CPP amplitude measurement was chosen a priori based on the interpretation of the motor potential inflection point as the beginning of response initiation and thus the point of commitment to a decision alternative, but it should be noted that this allows for no time delay between evidence accumulation and its downstream impact on motor preparation/execution (see Supplementary Fig. 4), and is considerably delayed with respect to EMG initiation (mean −135 ms) and the time window chosen in previous studies[27], underlining the importance of our repeating the tests for neighboring time windows. These analyses were repeated using the untransformed ERP waveforms without subtracting the auditory evoked

component, and results remained qualitatively unchanged (see Supplementary Fig. 3). To assess whether the CPP amplitude at response was significantly greater than zero when subjects made incorrect decisions, we pooled error trials across experimental conditions and computed a repeated-measures ANOVA across subjects. The peak time of the mean CPP within each subject, evidence-level and Speed/Accuracy regime was measured by finding the maximum amplitude between −150 ms and +100 ms in a smoothed average CPP trace. In order to minimize the spurious influence of instantaneous measurement noise in the peak time identification, we smoothed the CPP traces for these measurements by applying local regression (LOWESS) in a moving window of 200 ms. A 2-Way ANOVA was computed to determine the effect of Speed/Accuracy regime and evidence-strength on this delay.

**Neural signatures of motor preparation.** Motor preparation signals were measured in the decrease of Mu/Beta amplitude (8–30 Hz; integrated across both bands as in[29,35]) at motor cortical sites "C3" (left) and "C4" (right) for the preparation of contralateral responses. Spectral amplitude was quantified using a standard short-time Fourier transform with a boxcar window size of 300 ms at intervals of 50 ms. Motor preparation at baseline activation and at response were quantified as the Mu/Beta amplitude in the 300 ms preceding evidence onset and the button click, respectively, separately for the hemisphere contralateral and ipsilateral to the eventually executed response on a single-trial basis. To allow for relationships with RT, all measures were statistically assessed via linear mixed-effects models (see Statistical analysis section above). Excluding the SSVEP frequencies (20 Hz and 25 Hz) from the Mu/Beta computations did not change the pattern of results. The rate of change in motor preparation was calculated on a single-trial basis by measuring the slope of a line fit to the Mu/Beta amplitude in the interval between −350 and −150 ms relative to the response, chosen to capture as long a section of decision formation as is feasible while avoiding influences of post-response changes due to temporal blur associated with the 300-ms windows. A repeated-measures ANOVA was computed to test the significance of the influence of stimulus Contrast level and Speed/Accuracy emphasis on the Mu/Beta slope contralateral to response for trials that resulted in a correct response only. Restricting this analysis to correct trials ensured a positive relationship between the physical evidence and the neural measure of motor preparation.

To follow up on the finding of a CPP amplitude decline over RT, we performed an additional test for a dynamic urgency component at the level of motor preparation by examining Mu/Beta amplitude ipsilateral to the correct responses. If the dynamic urgency component is sufficiently strong, then its level at response may increase with increasing RT. To test this we computed a linear mixed-effects model on the excursion of ipsilateral Mu/Beta, computed as the difference between levels at response and at the pre-evidence baseline. It was important to measure excursion in this case in order to account for the significant variation of baseline motor preparation with RT, which otherwise could obscure the influence of systematic urgency increases during evidence accumulation.

Note that an alternative way to measure motor preparation is provided in the lateralised readiness potential (LRP), which has been previously used to finely trace differential decision formation dynamics[28,53,68,69]. The LRP is a differential signal reflecting the relative motor preparation for the chosen versus unchosen alternative, and is thus less suited to our current purposes of examining the influence of urgency signals, which commonly activate both alternative motor signals.

We have previously demonstrated that trial-to-trial variability in the dynamics of decision-related activity at multiple sensorimotor levels predicts RT[27,28]. To examine whether the current 2-alternative paradigm produces similar relationships, we plotted motor preparation, evidence accumulation, and sensory evidence encoding waveforms for equal-sized fast, medium, and slow RT bins (Supplementary Fig. 6). We computed within-subject correlations between the rate of rise of both the abstract and effector-selective (for the executed response) decision signals and the mean RT across the three RT bins. In the Results, we report the inverse of these slope measures, so that greater positive values indicate a steeper build-up in motor preparation and a positive correlation between inverse slopes and RT indicate that faster responses occurred on trials with a more rapid increase in motor preparation.

**EMG signatures of response execution.** EMG data were analyzed for effects on movement onset times and muscle activation levels. Motor time was quantified on a single-trial basis as the time between the onset of the muscle activity burst closest to response and the registration of a button click. EMG onset bursts were identified manually by visual inspection of the raw data, using a custom-made Graphical User Interface, and the results were verified on times derived from an automated algorithm relying on changes in variance of the broadband EMG signal and on the times estimated by another, independent human observer. The difference in the mean delay between the onset of muscle activation and the button click ("motor time") between Speed/Accuracy regimes was assessed using a two-tailed t-test. We further measured the number of muscle onset bursts (partial errors) in the response-withholding thumb. Differences between Speed and Accuracy emphasis were evaluated using a paired t-test. The effect of stimulus Contrast and Speed/Accuracy regime on the mean delay between EMG burst onset and the peak of the CPP was evaluated by a 2-Way repeated-measures ANOVA. Muscle activation was

quantified as the mean spectral amplitude between 10 and 250 Hz in 100-ms time windows stepped by 25 ms. The ultimate response-producing muscle activation was quantified as the spectral EMG amplitude in the responding thumb in the 100 ms preceding the button click. Insight into evidence-independent components of muscle activation was sought by quantifying the mean EMG spectral amplitude in the response-withholding thumb in a 100-ms time window preceding the mean onset time of the response-producing EMG burst (−225 ms to −125 ms). In this latter analysis only correct trials were included so that the sensory evidence runs counter to the measured action alternative, enabling us to more confidently attribute any increase over RT to evidence-independent urgency. Both measures were evaluated by linear mixed effect models.

On a single-trial basis, we additionally measured the rate of building muscle activation during responses initiation. Specifically, we measured the slope of a line fit to the spectral muscle activation timecourse (as before but stepped by 5 ms for increased resolution) in the response-executing thumb in the interval between −175 ms and −125 ms relative to the button click. This interval was chosen based on visual inspection of grand-average traces in single subjects. We statistically tested these temporal EMG slope measures using linear mixed effect models.

**Pupillometry**. To examine the role of pupil-linked arousal systems in the speed pressure adaptations, we continuously measured pupil size using the eye tracker. Pupil size was compared across Speed/Accuracy regimes in the 500 ms preceding evidence onset by a *t*-test. To test the influence of time on pupil size, we computed mean pupil size in 30 50-ms time windows spaced at 50 ms starting at stimulus onset. We then tested these time series for a significant interaction between Speed/Accuracy emphasis and Time across subjects using a 2-Way ($2 \times 30$), repeated-measures ANOVA. In order to test whether, above and beyond the average adjustments for speed pressure, variations in pupil size were linked with variations in differential evidence representation, we split the trials in each individual condition into two pupil-size bins based on mean pupil size between 0 and 1500 ms after evidence onset, using a broad time window to account for the protracted pupil response even to short, transient stimuli (see e.g., Wang et al.[70]). We then computed a 4-Way ANOVA including the factors Speed/Accuracy Regime, Contrast, Target type (Left/Right), and Pupil to capture the effect of pupil size on the differential SSVEP in a Pupil × Target type interaction. Here differential SSVEP was measured in the same time frame during which the Speed/Accuracy effect was significant (350–550 ms).

**Code availability**. Matlab code used in this project for stimulus presentation and analysis is available at https://osf.io/ksy5n/.

## Data availability
The data that support the findings of this study are available at https://osf.io/ksy5n/.

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

## Acknowledgements

This study was supported by grants from the United States National Science Foundation (BCS-1358955 to S.P.K. and R.G.O'C.), the European Research Council (63829 to R.G. O'C.), and the European Recovery Program-fellowship by the German National Merit Foundation (to N.A.S.). The authors are grateful for substantial feedback they received on the manuscript from Isabel Vanegas, Peter Murphy, Ariel Zylberberg, Dave McGovern, and Ger Loughnane.

## Author contributions

N.A.S., R.G.O'C. and S.P.K. designed the experiments. N.A.S. conducted the experiments and analyzed the data. N.A.S., R.G.O'C. and S.P.K. wrote the paper.

## Additional information

**Competing interests:** The authors declare no competing interests.

