## [Peer Review File · Nature Communications]

Reviewers' comments:

Reviewer #1 (Remarks to the Author):

Summary

In this paper, the authors compare and combine different measures of the evidence accumulation process, and examine how those are affected by a speed-accuracy trade-off manipulation. They find that several neural measures at different levels of the decision making process reflect speed-accuracy judgments.

Main points:

Overall this is an interesting paper, and I very much applaud the effort of combining various measures that have previously been associated with evidence accumulation. However, at the same time I am not entirely clear on what the hypotheses to be tested are and what the model is of evidence accumulation that is being tested. I am afraid that this lack of clear and integrative message limits the influence of this paper on the field.

In addition, I have several more specific points:

- 1) It is unclear what the role is of CAFs in the paper since those are virtually not discussed in the text
- 2) Previous studies (Gluth et al., 2013; van Vugt et al., 2014) have related the Lateralized Readiness Potential to motor preparation as well. Given that you discuss motor preparation (and you index it by beta oscillations) it is curious that you leave those earlier papers out of the discussion.
- 3) To complete the story about the various neural measures associated with evidence accumulation, it would be relevant to know which ones can predict response time.
- 4) I think it is great you are using LMEs to examine the data. However, in the tables it is not always clear what the model was that was fitted. It would be good to include the actual equation in the table caption.
- 5) In part of the paper, anovas are used, and in another part, LMEs. Why did you make this switch? Moreover, to demonstrate that a factor is significant in LMEs, you need to do an anova comparing the model with and without the relevant factor. The t-statistic is only a test of the magnitude of the relevant regression coefficient.
- 6) At several points (e.g., p. 6 "Accuracy regime did not improve the fit significantly ($\Delta \log \text{likelihood} = 1.8236$; $p = 0.24$). Thus, response accuracy declined over RT in the Accuracy as much as the Speed regime.") the failure to reach significance is interpreted to mean equivalence. However, to prove equivalence you need to do an equivalence test or compute a Bayes Factor.
- 7) I think the supplementary tables are a bit hard to parse. Moreover, I do not see how supplementary table 2d-e showed that "This modulation was transient, emerging following evidence onset and lasting until just before the response" (p.10).

It would be awesome if the authors could also share their data (in the interest of transparency and openness of science)

Minor points:

- 1) "but was curtailed in the level it reached at response by the downstream, motorlevel urgency." What does this mean? How do you know it was curtailed?
- 2) It is not clear what the rationale is for the particular decision task chosen. In addition, what is the reason for the fading in of the stimuli (rather than presenting them in one go).
- 3) How did the algorithm for removing the auditory stimulus-evoked activity work?
- 4) What is the purpose for computing "smooth traces" of the CPP?
- 5) p. 3 "A "static" component of urgency has been widely observed in raised baseline activity before evidence presentation" It seems to me that this would more likely reflect a bias rather than urgency.
- 6) At what time was the pupil measurement taken that is tested at the bottom of p. 12?
- 7) Figure 5 is a bit hard to parse, because it is not entirely clear what the different arrows between the levels reflect.
- 8) On p.20, comparing the speed and accuracy emphasis in the EMG burst, it would be good to mention the mean EMG burst in speed vs accuracy, because the difference seems quite small.
- 9) In suppl figure 2 it would be helpful to indicate task events such as stimulus and response with vertical lines to clarify what the various peaks mean.
- 10) Suppl figure 5 is very difficult to parse because too many different things are plotting in one figure.

Reviewer #2 (Remarks to the Author):

Using a "multi-level recording paradigm" in which many (electro)physiological signals are measured concurrently, the authors aim to answer the question during what stage of a perceptual choice speed pressure is applied. They argue that speed pressure may be best understood as a stimulus-independent signal that is summed to motor preparatory signals (i.e., after an evidence accumulation stage).

While this is an interesting as well as important hypothesis, I don't believe the current manuscript provides strong evidence for it: I worry about the statistical analyses, which preclude me to fully endorse this paper in its current form. Moreover, I found the manuscript quite difficult to follow at times, partly because I am not an expert on every technique applied in this study, but partly also because the writing is quite dense, in particular in the results sections (related to the reporting of the results), and sometimes there were confusing mistakes.

Specific points:

There are many regression analyses reported in this paper, increasing the chance of finding significant spurious effects (type II errors). It would have been good to include an appropriate correction on the Type II error rate to limit these chances. Given the current interest in replicable results, I think this is especially important.

Although there are many regressions reported, it was not clear to me why these specifically

where selected. For example, the specific choice for a random effects structure (only subject intercepts as far as I could tell) remained unmotivated, even though this is not considered the optimal way of including random effects. Similarly, the in- or exclusion of certain predictors seemed slightly adhoc (although to some extent consistent). Typically, model comparisons are reported to argue for or against including certain factors.

Moving part of the details of the regression analyses and ANOVAs to the supplements is not a good choice, in my opinion. For me it became very difficult to follow the reasoning in the main text.

On page 8 the authors study the accuracy decline per unit time by doing a linear regression over all RTs *after* a participants' RT bin of maximum performance (or the mean RT in that bin). This analysis seems problematic, since the results depend on (a) The number of RT bins (which influences the estimation of the point of maximum performance and (b) the type of pooling of the data. It was not clear to me whether the pooling was done over Speed and Accuracy regimes, which seems to be problematic for detecting e.g. differences in the slope of the regression lines per speed/accuracy regime.

On page 13, the authors report a repeated measures ANOVA to study how Speed/Accuracy regime predicts pupil size. The repeated measures in this analyses are the time points at which the pupils size is determined. However, since these are not independent, estimates of the effect of the regimes are inflated (i.e., the standard errors are reduced), potentially leading to incorrect conclusions.

Smaller points:

Page 3: "collapsing bound" is mentioned, but the uninformed reader might not know this concept.

Page 8: It was not clear to me what the dependent variable was in the CAF regression analysis (also discussed above). Was this accuracy (correct/incorrect), or *proportion* correct per RT bin?

Fig 3. "traces are baseline-corrected to the 500ms..." Does this mean this exact time point, or an average in the [500,0] ms interval?

Page 15: a p-value of $p=0.09$ cannot be considered a trend.

Page 41: "accuracy deadline". It is not clear to me what the duration is that the evidence was on the screen. I think it 2.4 s, based on Fig 1a, but this could be specified.

Reviewer #3 (Remarks to the Author):

This manuscript describes a human electroencephalography (EEG) study which aims at characterizing how urgency affects perceptual decision-making at sensory, decision and motor processing stages. Their specific research question concerns whether urgency is applied "globally" at all processing stages, or whether distinct adjustments take place at different processing stages - a question grounded in previous, sometimes contradictory

work on urgency. For this purpose, the authors use a well-controlled perceptual discrimination paradigm in which either the accuracy (low urgency) or the speed (high urgency) of performed decisions is emphasized through changing payoff strategies - as in previous work on this question. The authors show that urgency is reflected as a static increase in motor preparation (both cortical and muscular) even before the decision evidence is presented, but not reflected upstream in the neural correlates of decision evidence accumulation. Furthermore, the authors report that the sensory processing of decision evidence is enhanced under high urgency to reduce the performance loss due to the shift in speed-accuracy tradeoff under temporal pressure.

I found the manuscript to be an interesting read, although difficult to follow at times, based on a well-designed paradigm and clever analyses of the recorded EEG data to decompose perceptual decision-making into sensory signals (steady-state visual evoked responses), decision evidence accumulation signals (the centro-parietal positivity component identified by the authors in earlier work) and motor preparation signals (band-limited mu/beta power overlying motor cortex, and electromyographic activity in thumbs used to report decisions). The pattern of findings reported in the manuscript is however complex and seemingly paradoxical at first glance, and thus bound to be fully grasped only by specialists. Therefore, at this stage, I am not entirely convinced that the obtained findings provide a sufficient advance in the existing knowledge about speed-accuracy trade-offs and urgency to justify publication in a general-audience journal - although I would be happy to revise my judgment if the authors can increase the generality of the findings and their appeal to a non-specialist audience in a subsequent revision of the manuscript.

I have several comments, listed below, which I think the authors should consider for revising their manuscript. Some of these comments reflect issues that I have with reconciling the different, seemingly contradictory findings presented by the authors in the manuscript. Other reflect the incomplete description of the analyses applied to the data (e.g., the temporal windows of interest used for many of the statistical tests), which render their assessment very difficult - especially the use of mixed-effects models which appear unnecessary given the within-subjects design used by the authors.

Major comments:

* Introduction (p. 4): The authors draw a contrasted picture of the past literature on the effects of temporal pressure on the sensory processing of evidence during decision-making, suggesting that variations in the particular paradigms used to modulate urgency and in the neuroimaging modalities used to study it (animal electrophysiology, human fMRI, MEG/EEG) might explain some of the observed inconsistencies. Which such a line of reasoning can indeed explain the important differences observed across studies reported by the authors, it also suggests that the findings reported by the authors in the present study might be peculiar to the particular paradigm they used and the particular neuroimaging modality (EEG) they relied on. The authors should provide arguments as to why and how their findings (in particular the sensory enhancement of the differential evidence observed under urgency) are expected to generalize to other stimuli and manipulations of urgency.

* Abstract (p. 2) and Introduction (p. 7): Since the representation of cumulative evidence is found downstream from the sensory processing of the evidence samples, it is hard to understand from the abstract how to reconcile an enhanced sensory representation of the differential evidence (upstream) with a weaker representation of the cumulative evidence (downstream). A standard prediction would be that because of the increased representation of the differential evidence under temporal pressure, the ramping up of the cumulative evidence should be increased and thus reach the same level at motor response onset as in the non-speeded condition. The authors might want to re-write these sections in such a way that does not trigger such apparent inconsistency - which is later considered and discussed by the authors.

* Results (p. 10): The authors often do not describe precisely the time window at which differential evidence is "boosted under speed pressure". This is a general comment about the reporting of findings throughout the Results section, where important details are often missing. Also, the authors make a crucial claim about the transient nature of this effect, restricted to the evidence accumulation period, but mention no statistics to support this claim, no illustration of this effect and refer merely to a supplementary table. Because this sensory enhancement under speed pressure is one of the most surprising (and novel) effects reported in the manuscript, the authors should in my opinion provide all the necessary information to describe this effect.

* Results (p. 11): I do not understand the motivation underlying the use of a mixed-effects model throughout the Results section. Indeed, the full experimental design appears to be using only within-subjects factors, which affords the use of classical repeated-measures ANOVAs and does not require the use of mixed-effects models. The authors never justify the use of a mixed-effects models, neither in the Results nor in the Methods sections, something which is annoying given that such a statistical model does not seem fully warranted given their experimental design (at least as far as I can tell). Another concern is whether the reported statistics from the mixed-effects model are random-effects statistics (which take the variability across tested subjects as measure of dispersion) or fixed-effects statistics (which take the variability within tested subjects as measure of dispersion). Again, this important point is not described anywhere in the Methods or even Supplementary Material sections. I therefore cannot assess the validity of the statistical analyses relying on mixed-effects models, and remain doubtful regarding the strength of the reported effects based on such models.

* Results (p. 12): As indicated earlier, the authors do not report the latency at which they observed pupil effects, and described the findings too succinctly to assess the strength of the reported effect.

* Results (p. 17, "The decreased peri-response amplitude [...] directly at the CPP level." I am unconvinced that narrowing the bounds is necessarily "dynamic" per se. In my opinion, it can be a static effect that is dissociable from the baseline increase in motor activity. Also, this effect seems to be present for both speed and accuracy conditions - when I was expecting to see it only in the speed (urgency) condition. Can the authors elaborate on this theoretical point, and explain why their results truly support a dynamic view rather than a

static change in decision bound?

* Figures: Wherever possible, the authors should display error bars (e.g., s.e.m.) rather than only the mean across tested subjects. I can understand that individual subject data does not need to be plotted to assess the significance of an effect, but a measure of between-subject variability such as the s.e.m. is desperately needed to be shown for readers to assess (at least qualitatively) the strength of the reported findings.

* Results (p. 20, "responses are initiated while the evidence accumulation is ongoing"): The authors should avoid making too clear/strong reverse inference here. They should reformulate the cited statement as "[...] while the neural correlates of evidence accumulation are still present", especially because the statement is found in the Results section, not in the Discussion section where such reverse inference might be more appropriate.

Minor comments:

* Introduction (pp. 5-6, "Resolving all of these questions [...] into motor preparation."): I found this paragraph which describes the different EEG measures of sensory processing, evidence accumulation and response preparation to be too long. The authors should in my opinion shorten this paragraph.

* Results (p. 7): I am unclear as to how a "blocked" urgency design across trials could not have afforded to examine the effects of urgency at each neural processing level - especially because the identification of these different neural processing levels is done based on a "a priori" definition of particular neural signals that seems entirely unrelated to the "blocked" or "interleaved" nature of the experimental design. Can the authors either reformulate or remove this statement?

* Figure 1: The auditory cue should be shown as one line on panel (a). It is a very important feature of the paradigm which is apparently missing from the figure.

* Discussion (p. 27): It is unclear what the authors mean by "collapsing decision bound": do the authors mean a collapsing bound over time within a trial (which I don't think there is any evidence for in the reported findings, but which is what is meant by collapsing bound in computational modeling papers), or simply an overall lower decision bound which does not shrink over time within each trial?

Reviewers' comments:

Reviewer #1 (Remarks to the Author):

Summary

In this paper, the authors compare and combine different measures of the evidence accumulation process, and examine how those are affected by a speed-accuracy trade-off manipulation. They find that several neural measures at different levels of the decision making process reflect speed-accuracy judgments.

Main points:

Overall this is an interesting paper, and I very much applaud the effort of combining various measures that have previously been associated with evidence accumulation. However, at the same time I am not entirely clear on what the hypotheses to be tested are and what the model is of evidence accumulation that is being tested. I am afraid that this lack of clear and integrative message limits the influence of this paper on the field.

We appreciate the reviewer's balanced feedback. With the guidance of these comments and those of the other reviewers, we have been able to significantly improve the clarity of our hypotheses and better integrate our findings to convey a clear message on the broader implications of the study.

To briefly presage this clarified message here, neurophysiological investigations of the speed-accuracy tradeoff have centred on the role of evidence-independent 'urgency signals' in motor preparation circuits, which serve to expedite decision commitment and action execution at the expense of accuracy. But whether and how neural decision formation dynamics are adjusted to account for speed pressure at processing levels other than motor preparation remains unclear. One emerging hypothesis states that urgency signals arise from a global gain enhancement mediated by neuromodulatory arousal systems, which would predict that qualitatively similar changes in neural activity would occur at all levels of the sensorimotor hierarchy. Our human decision paradigm provided a unique opportunity to test this, through the parallel tracing of sensory encoding, evidence accumulation, motor preparation, action execution and arousal. Our results showed that speed pressure did indeed impact on multiple processing levels but in crucially distinct ways. Evidence-independent urgency signals were applied at the level of motor preparation and reflected in downstream muscle activation, but were not applied at upstream processing levels (sensory encoding and evidence accumulation). Instead, speed pressure rendered the representations of the two alternatives more distinguishable at the sensory level, which had a positive effect on response accuracy that partially countered the accuracy cost caused by the more dominant motor-level urgency effects. These opponent speed-pressure adjustments at the sensory and motor levels had separate knock-on effects on the buildup rate and pre-response amplitude, respectively, of an intermediate, motor-independent representation of cumulative evidence.

Thus, our study reveals the multi-faceted nature of the brain's adjustments to speed pressure over several levels of the sensorimotor pathway. These findings have direct and important implications for models of decision making which, to date, have predominantly suggested that the only adjustment the brain makes to take account of time pressure in decision making is the lowering of the decision bounds. Our data point to critical additional adjustments that are not accommodated in current models (enhanced evidence encoding, faster evidence accumulation, faster action execution). Further, our data are not only consistent with recent models centered on a bounded decision variable driven by a combination of cumulative evidence plus an urgency component (e.g. Hanks et al., 2014; Murphy et al., 2016), but go further in showing that the brain implements this scheme using two decision layers: one encoding cumulative evidence on its own (the abstract level of the CPP) and the other, motor layer adding the urgency component and imposing the ultimate action-triggering threshold. This is particularly relevant given the recent increased scrutiny of brain areas involved in motor preparation (e.g. LIP) which appear to reflect, but not necessarily compute, the evidence accumulation process (e.g. Katz et al., 2016, Nature), and the subsequent proposals that such areas may receive a feed of cumulative evidence from an upstream process (e.g. Pisupati, Chartarifsky and Churchland, 2016, TiCS).

We have reworked the abstract, introduction and discussion to ensure that this message is conveyed more clearly.

In addition, I have several more specific points:

1) It is unclear what the role is of CAFs in the paper since those are virtually not discussed in the text

We thank the reviewer for pointing out this lack of clarity and have revised the way we introduce the conditional accuracy functions in the Results (see page 7) and the Methods (page 34) section. The principal reason for including CAFs in this manuscript was to demonstrate one of the key behavioural indicators of dynamic urgency signals or, equivalently, collapsing bounds: a progressive reduction in accuracy as a function of increasing reaction time, consistent with less cumulative evidence being required in order to trigger commitment as time elapses. In addition, the low response accuracy for the very earliest responses is consistent with inter-trial variability in the starting point of the evidence accumulation process which we subsequently link to variations in the baseline activity of motor preparation signals (page 13).

2) Previous studies (Gluth et al., 2013; van Vugt et al., 2014) have related the Lateralized Readiness Potential to motor preparation as well. Given that you discuss motor preparation (and you index it by beta oscillations) it is curious that you leave those earlier papers out of the discussion.

The reviewer correctly points out that the LRP signal is an alternative index of motor preparation, and cites two excellent examples of its effective use in studying decision making. Indeed, previous work, including our own (Kelly and O'Connell, 2013), has demonstrated that the LRP also exhibits

evidence accumulation dynamics. Here, however, the LRP was not best suited to addressing our specific hypotheses because it is a differential signal reflecting the relative motor preparation for the chosen versus unchosen alternative. In the absence of any prior information, urgency signals should be equal in magnitude on average for both choice alternatives and therefore would be cancelled out in the calculation of the LRP (subtraction of contralateral from ipsilateral ERPs and averaging across left and right responses). In contrast, our Mu/Beta signals, just like the EMG signals recorded on separate thumbs, provide separate indices of motor preparation for each decision alternative, and thus the ability to identify urgency effects as a common component of activation in both alternatives.

We thank the reviewer for highlighting the need to make these considerations more apparent to the reader. We have now added this rationale behind choosing Mu/Beta over LRP as an index of motor preparation in the methods section (page 44), and cite both of the papers mentioned by the reviewer.

3) To complete the story about the various neural measures associated with evidence accumulation, it would be relevant to know which ones can predict response time.

We agree with the reviewer that a relationship between neural signals and RT is a valuable indicator of their functional importance. In previous papers we have demonstrated that the buildup of both the CPP and motor preparation signals predicts RT across a number of different tasks (e.g. O'Connell et al., 2012; Kelly and O'Connell., 2013; Loughnane et al., 2016). It is worthwhile to verify that the same relationships hold for the present tasks and we have hence added a figure (Supplementary Fig. 6) in which we display our neural indices of sensory representation (SSVEP), evidence accumulation (CPP) and motor preparation (Mu and Beta amplitude) split by reaction time. Statistical comparisons revealed a significant, negative relationship between RT and the build-up rates of both the evidence accumulation ($t(15) = -4.67$; $p = 0.00030$) and motor preparation signals ($t(15) = -6.46$; $p = 1.08e-5$). In contrast with our previous study of a continuous monitoring task (O'Connell et al., 2012), we did not find a significant relationship between RT and SSVEP amplitude in this discrete task (see Supplementary Figure 6, reproduced below).

Supplementary Figure 6: Time course of sensory encoding, accumulation and motor preparation as a function of RT

For each individual subject and condition (Regime, Contrast, onset delay and Target Type), trials were split into reaction time tertiles and each tertile was then collapsed across conditions. Mean neural signals of the three RT bins are plotted locked to evidence onset (a, c, e) and response (b, d, f). Vertical lines in the stimulus-locked panels indicate mean RT per tertile. For each time point, we computed the correlation between RT (tertile means) and mean signal amplitude within subjects. We then computed t-tests to test whether the distributions were significantly different from zero across subjects. Time points with significant correlations are marked with gray stars at the bottom of each panel. (a) Stimulus-locked Mu/Beta signals predict reaction time shortly before and after evidence onset with greater motor preparation (lower Mu/Beta amplitude) predicting faster response

times. Significant negative correlations between motor preparation and RT more than 1000ms after evidence onset are a byproduct of a decrease in motor preparation after response execution. (b) Response-locked motor preparation has a negative relationship with RT until around 300 ms before the response, reflecting the fact that motor preparation builds over a narrower timeframe on the trials with faster RT, and then reaches uniform levels at the time of response execution. (c) Stimulus-locked CPP predicts RT between 200 and 600 ms after evidence onset, consistent with shallower evidence integration on trials with slower responses. Similar to motor preparation, positive correlations between CPP amplitude and RT beyond about 700 ms result from the decrease in signal amplitude once a response is made. (d) As expected, response-locked CPP traces show positive correlations with RT well in advance of the response and negative correlations after response commitment. Replicating the results of trials split by condition (Figure 4h), CPP amplitude is lower for very fast and very slow responses around the time of decision commitment (-130 to -70 ms). (e) Stimulus-locked SSVEP does not predict RT at any time point before the mean reaction time of the fastest RT bin. (f) Response-locked SSVEP amplitude correlates positively with RT until 350 ms before the response. This is simply due to the initial ramp-up of the SSVEP differential at evidence onset occurring at different pre-response times for the different RT bins.

4) I think it is great you are using LMEs to examine the data. However, in the tables it is not always clear what the model was that was fitted. It would be good to include the actual equation in the table caption.

We have now included the equations for all regression models in a new section of the supplementary information that exhaustively lays out all results of each statistical test, including the steps taken to determine the most appropriate final model in each case (see next reply).

5) In part of the paper, anovas are used, and in another part, LMEs. Why did you make this switch? Moreover, to demonstrate that a factor is significant in LMEs, you need to do an anova comparing the model with and without the relevant factor. The t-statistic is only a test of the magnitude of the relevant regression coefficient.

We are grateful to the reviewer for pointing out the need to clarify choices of statistical test, and the appropriate test to use for demonstrating the significance of each factor, both of which we have now remedied. As we now explain fully in a new, dedicated statistical analysis section of the Methods (see page 36):

For all statistical tests involving multiple independent variables, we computed linear mixed-effects models on single-trial data if reaction time (RT) was a relevant independent variable, and otherwise used repeated-measures analyses of variance (ANOVA) on subject-averaged data. The use of linear mixed-effects models with the factor of RT was necessary to disentangle effects of RT arising from time-dependent influences within a trial (e.g. dynamic urgency) from other experimental factors that affected RT (e.g.,

Contrast, Speed/Accuracy Regime). To take a hypothetical example, if an identical collapsing bound were applied in the Speed and Accuracy Regime and it was through some mechanism other than bound adjustment that RTs were longer in the Accuracy Regime, the fact that the bound would have collapsed further with more elapsed time would mean that a basic comparison of an accumulator signal amplitude across Regimes would exhibit a difference, leading to an RT effect that masquerades as an amplitude (bound) effect. Controlling for such factors requires inclusion of the absolute value of RT as an independent variable rather than its rank within condition (e.g. quantile 1, 2, etc), necessitating the use of linear mixed effects models.

In laying out this rationale explicitly we in fact discovered two instances where an LME approach was originally applied but was not necessary: 1) We now examine the relationship between SSVEP amplitude and response accuracy using a 3-Way repeated-measures ANOVA in the revised manuscript (see page 10). 2) To evaluate whether the decrease in response accuracy over RT for slow responses is steeper under Speed pressure, we now fit lines to this part of the CAF within each subject and condition and compute a 3-Way repeated-measures ANOVA to test for an effect of Speed/Accuracy Regime (page 7). The results of both analyses remain qualitatively the same.

Further, we have revised how LME models are constructed and their results stated, in line with the comments of Reviewers 1 and 2. We had originally quoted t-test statistics to address the significant influence of each fixed effects factor in each model, but both Reviewer 1 and Reviewer 2 have pointed out that to use LMEs properly, each factor needs to be separately tested for its significant contribution toward a good fit to the data. We have thus switched to this approach throughout the revised manuscript, now quoting chi-squared statistics for model comparisons testing for a significant contribution of each factor to the overall fit of the model. Based on comment #2 from Reviewer 2, we have also revised our approach to the inclusion of random effects factors. Whereas originally we included only random intercepts across subjects, we now perform an iterative procedure to determine for each test which random slopes factors contribute significantly to the model fit. We explicitly state this strategy for statistical testing in the new *Statistical analysis* section of the Methods (page 36), and provide an exhaustive account of every step of the statistical model refinement process described above in a new section "*Supplementary statistical analysis details*" in the Supplementary Materials.

6) At several points (e.g., p. 6 "Accuracy regime did not improve the fit significantly ($\Delta \log \text{likelihood} = 1.8236$; $p = 0.24$). Thus, response accuracy declined over RT in the Accuracy as much as the Speed regime.") the failure to reach significance is interpreted to mean equivalence. However, to prove equivalence you need to do an equivalence test or compute a Bayes Factor.

We thank the reviewer for raising this issue. We have revised this analysis (see previous response) and now compute a 3-Way repeated-measures ANOVA. We follow this up by comparing the best model fit to the data with a model that additionally considers Speed/Accuracy emphasis. We now explicitly state that the evidence supporting an identical decrease in

performance is moderate ($BF_{01} = 2.197$; see page 7). For the purposes of illustrating how the major speed pressure adjustment effects fit together in our schematic (Fig 5), we assume no difference in the rate of urgency buildup, but exact equivalence is not central to the point we are making and we take care not to strongly assume it (see page 19 and 27).

We have also computed Bayes Factors in three other instances in which we obtained a non-significant result. In the case of the lack of main effects of any experimental factor on CPP amplitude at baseline (all $BF_{01} > 33$, page 14) and on Mu/Beta amplitude at response (all $BF_{01} > 12$, page 14), the Bayes Factors indicate strong evidence for the null-hypothesis, whereas the absence of a main effect of RT on Mu/Beta excursion for the withheld response is less conclusive ($BF_{01} = 1.38$; page 21).

7) I think the supplementary tables are a bit hard to parse. Moreover, I do not see how supplementary table 2d-e showed that "This modulation was transient, emerging following evidence onset and lasting until just before the response" (p.10).

As mentioned above, we acted upon all of the reviewers' feedback regarding clarity of statistical analyses by including a new statistical analysis section in the main text (Methods) and making a revised Supplementary section exhaustively detailing all statistical model steps and outcomes. The specific instance pointed to by the reviewer was indeed particularly unclear and we have now made it explicit (See page 9). We now specify that the Regime (Speed vs Accuracy) x Target Type (Left vs Right) interaction effect which captures the sensory evidence boost under speed pressure was significant immediately prior to response execution but not after response execution. These are the tests previously referred to in the Supplementary Table 2e-d and now in Supplementary Table 2e-f. We thank the reviewer for pointing out the lack of clarity.

It would be awesome if the authors could also share their data (in the interest of transparency and openness of science)

We agree wholeheartedly; in fact we stated our commitment in our grant from the National Science Foundation, to share all data and analysis code immediately upon acceptance of each paper for publication.

Minor points:

1) "but was curtailed in the level it reached at response by the downstream, motorlevel urgency." What does this mean? How do you know it was curtailed?

We have now revised our summary of results in the introduction guided by all reviewer comments and no longer use the term in that section (Page 6). What we meant was that the action execution threshold is set at the motor level, and as the time of this threshold-crossing is dictated partly by urgency components, urgency determines the level of cumulative evidence reached at response (less cumulative evidence under greater urgency).

2) It is not clear what the rationale is for the particular decision task chosen. In addition, what is the reason for the fading in of the stimuli (rather than presenting them in one go).

We used this contrast discrimination task because it allowed us to non-invasively isolate neural signals that specifically reflect the sensory input to the decision process using the Steady-State Visual-Evoked Potential (SSVEP) technique. SSVEP amplitude varies monotonically with stimulus contrast, and therefore if contrast is what is being decided on, we gain access to the evidence representation directly through SSVEP amplitudes measured noninvasively. Trials began with a linear fade-in of the grating stimulus in order to eliminate transient visual-evoked potentials generated by sudden onsets, which can interfere with the read-out of other broad-band EEG components like the centro-parietal positivity (O'Connell et al., 2012). We have added such details and reasoning to the section of the Methods that describes our task design (Page 7).

3) How did the algorithm for removing the auditory stimulus-evoked activity work?

We have added further details on this method to the section 'Evidence accumulation' in the Methods (Page 41), which now describes how the algorithm exploits the variability in response latency across single trials to iteratively fit a stimulus-locked and a response-locked component to the data. These components are derived iteratively based on the discrepancy between a superposition of stimulus- and response-locked components on each individual trial and the actual data. Critically, the algorithm is naive to all stimulus conditions, and the resulting stimulus-locked component is constrained to be invariant across all trials within each individual subject, so that differences in potentials evoked by stimulus contrast or Speed/Accuracy regime are left untouched by this method. The same stimulus-locked auditory evoked potential was then subtracted from the evoked potential of all trials.

4) What is the purpose for computing "smooth traces" of the CPP?

Other than for display purposes, we applied temporal smoothing only in the derivation of CPP peak latencies relative to response execution. Here, the smoothing technique was applied to reduce the impact of high frequency noise fluctuations on these measurements. Specifically, we applied locally weighted smoothing, which has been used in analyses of EEG (Hine et al., 2017), fMRI (Fair et al., 2008) and LFP (Wang et al., 2009) data. We have now clarified the description of this smoothing step on page 42.

Damien A. Fair, Alexander L. Cohen, Nico U. F. Dosenbach, Jessica A. Church, Francis M. Miezin, Deanna M. Barch, Marcus E. Raichle, Steven E. Petersen and Bradley L. Schlaggar (2008). "The maturing architecture of the brain's default network". PNAS, 105 (10) 4028-4032.

Gabriel Emile Hine, Emanuele Maiorana, Patrizio Campisi (2017). "Resting-state EEG: A Study on its non-Stationarity for Biometric Applications" (2017). International Conference of the Biometrics Special Interest Group (BIOSIG), Darmstadt, Germany

Zhisong Wang, Alexander Maier, Nikos K. Logothetis, and Hualou Liang (2009). "Extraction of Bistable-Percept Related Features From Local Field Potential by Integration of Local Regression and Common Spatial Patterns". IEEE TRANSACTIONS ON BIOMEDICAL ENGINEERING, 56(8)

5) p. 3 "A "static" component of urgency has been widely observed in raised baseline activity before evidence presentation" It seems to me that this would more likely reflect a bias rather than urgency.

The reviewer is correct that decision biases can manifest as shifts in baseline activity, as seen in several studies manipulating prior probability or value (e.g. Hanks et al., 2011, Rorie et al., 2010, de Lange et al., 2013). However, the critical distinction is that such biases are selective for one of the two alternatives (the more probable or valuable one in those studies). In contrast, the elevated baseline activity we observe under speed pressure is simultaneously in the preparation signals for both the executed and withheld response. One can view this as a simultaneous "bias" to respond for both hands (in a similar sense perhaps to a liberal criterion for reporting a signal in signal detection theory) but this would entail using the term "bias" in a different sense than is standard in research on 2-alternative decisions. In response to this important point, we have made clearer in the revision that "urgency" is defined as a common component of activation for both alternatives (see e.g. pages 3 and 13, Figure 5), as distinct from a selective one for one of them, as is the case for a bias.

6) At what time was the pupil measurement taken that is tested at the bottom of p. 12?

The baseline pupil diameter was measured as the mean pupil size between -500 to 0ms relative to evidence onset. In the analysis testing the relationship between pupil size and SSVEP amplitude, we measured pupil size as the mean pupil size between 0ms and 1500ms after stimulus-onset. We have modified this section of the Results to incorporate these important details (page 11).

7) Figure 5 is a bit hard to parse, because it is not entirely clear what the different arrows between the levels reflect.

We thank the reviewer for this valuable feedback. We have revised both the Figure and its caption to walk the reader through our interpretation of the experimental results with greater clarity. In particular, we have made a much clearer distinction between the arrows, using thick black ones to mark the two direct speed pressure adjustments, and thick grey ones to mark the knock-on effects of those adjustments at other hierarchical levels, and have removed the arrows that were unnecessary for making our point. We have also labelled the levels with panel letters to clearly point to each in the caption explanation, which we believe is now much improved in clarity (See Page 18, also pasted below for convenience).

Figure 5: Schematic summary of speed pressure effects across cortical sensorimotor levels. In our proposed architecture, neural sensory evidence signals (**d**) are temporally integrated at an intermediate, motor-independent level (**c**, representing the CPP), which in turn feeds **into the** motor-level buildup signal representing preparation for the alternative favored by the evidence (**b**, 'left' alternative in this example). At the motor level (**b**, representing Mu/Beta activity), this favored alternative races against the unfavored alternative (**b**, right), with the first one to cross the motor threshold determining the response choice and timing for that trial. An evidence-independent urgency signal (**a**) additively feeds both motor-level alternatives equally, and grows dynamically over time. The schematic illustrates idealized dynamics at all levels for a typical single trial (left-tilted, correct) in each Regime (Speed in red, Accuracy in blue), representing the primary, active speed pressure adjustments by **thick** black arrows, and their knock-on impact at other levels by **thick** gray arrows. **Speed pressure has two direct effects, one on the sensory (**d**) and one on the urgency signal (**a**).** At the sensory level (**d**), the differential representation of the increased versus decreased contrast grating is **greater under speed pressure than accuracy emphasis (red trace elevated over blue trace).** Note that this enhancement is not classed as an "urgency" influence because it is **not** evidence-independent - it acts to increase the differential specifically in the direction of the physical contrast difference. This sensory-level enhancement (**d**) leads to an increased rate of rise in the motor-independent accumulator signal (**c**, curved grey arrow) and is further reflected in downstream motor preparation (**b**, left, curved gray arrow). The second direct effect of speed pressure is an upward shift of the evidence-independent urgency component applied at the level of motor preparation. Based on the similar rates of decline in conditional accuracy for both Regimes (Figure 1c), we assume for parsimony that the rate of evidence-independent urgency buildup is the same for both Regimes, and that it is only the static component that is significantly adjusted under speed pressure in this paradigm. The addition of this growing urgency at the motor level (**b**, straight gray arrows) limits the amount of sensory evidence that can be accumulated before the motor threshold is crossed, resulting in an effective decrease in the attainable amplitude of the CPP over response time (light-shaded lines in **c**). Crucially, in this scheme the differences in the CPP arising from speed pressure need not require any speed pressure adjustment applied directly to that level of processing, but rather can be explained by the combination of two knock-on effects: while the sensory-level boost steepens the CPP's buildup so that its post-stimulus amplitude rises higher than for the Accuracy Regime, ultimately it is the motor-level threshold that triggers response execution, and the added urgency at that level tends to decrease the level that the CPP can reach by the time of the response. Note that the scales of the axes at different levels are not intended to be consistent, and the size of some effects are exaggerated to aid clarity of illustration.

8) On p.20, comparing the speed and accuracy emphasis in the EMG burst, it would be good to mention the mean EMG burst in speed vs accuracy, because the difference seems quite small.

We have added the mean motor times (Speed: -130.2 ms; Accuracy: -139.9 ms; difference: 9.7 +- 6.4 ms; $t(15)=6.1$, $p=2.07*10^{-5}$) and number of EMG bursts (Speed: $n_{burst} = 56.4 +- 29.5$; Accuracy: $n_{burst} = 42.3 +- 23.7$; $t(15)=4.6$, $p=0.00034$) and their respective standard deviations to the Results (see pages 20 and 21).

9) In suppl figure 2 it would be helpful to indicate task events such as stimulus and response with vertical lines to clarify what the various peaks mean.

As suggested, we have indicated stimulus onset and mean RT with vertical lines.

10) Suppl figure 5 is very difficult to parse because too many different things are plotting in one figure.

We have now separated the baseline amplitudes for the executed and withheld responses into two panels a and b, to make the different conditions easier to parse.

Reviewer #2 (Remarks to the Author):

Using a “multi-level recording paradigm” in which many (electro)physiological signals are measured concurrently, the authors aim to answer the question during what stage of a perceptual choice speed pressure is applied. They argue that speed pressure may be best understood as a stimulus-independent signal that is summed to motor preparatory signals (i.e., after an evidence accumulation stage).

While this is an interesting as well as important hypothesis, I don't believe the current manuscript provides strong evidence for it: I worry about the statistical analyses, which preclude me to fully endorse this paper in its current form. Moreover, I found the manuscript quite difficult to follow at times, partly because I am not an expert on every technique applied in this study, but partly also because the writing is quite dense, in particular in the results sections (related to the reporting of the results), and sometimes there were confusing mistakes.

We thank the reviewer for raising these concerns and guiding us to improve the written presentation and to explain our statistical approaches more carefully. Following the comments of all three reviewers, the manuscript has been substantially re-worked, with new dedicated sections added to make our statistical procedures explicitly clear, and we believe the manuscript is much improved as a result.

Below we have added numbers to the reviewer's comments for ease of reference elsewhere.

Specific points:

1) There are many regression analyses reported in this paper, increasing the chance of finding significant spurious effects (type II errors). It would have been good to include an appropriate correction on the Type II error rate to limit these chances. Given the current interest in replicable results, I think this is especially important.

The reviewer is raising a very important matter in light of the growing awareness of the systemic problems that have arisen in the scientific literature due to suboptimal practices in the reporting of statistical inferences. We assume that the reviewer means type I errors here (false positives), the risk of which increases with the number of independent tests conducted, and which is particularly problematic when each individual test has the potential to produce an important 'discovery'. In our paper however, while there were many tests carried out in total, our statistical analyses are actually centered on a relatively small number of primary tests, each designed to verify the central hypothesis that behavioural speed pressure adjustments are mediated by a single global 'urgency' influence that acts on neural activity at all processing levels. We then conducted a larger number of follow-up tests designed to verify the reliability of the outcome of each one of these primary tests. For example, in our test for such urgency effects at the sensory level, we found that instead of a main effect on SSVEP amplitude, the SSVEP *differential* was greater under speed pressure than under accuracy emphasis; an effect that is evident for both target directions and both difficulty levels (Fig. 2b). As a way to further verify the reliability of this result, we tested for manifestations of the same phenomenon in other dependent variables. If the observed SSVEP effect does indeed reflect an increase in differential sensory evidence under speed pressure, then we should observe faster rates of evidence accumulation (CPP build-up rate) and motor preparation (μ /beta build-up rate), and if it is consequential for behaviour then one would expect an association between SSVEP differential and choice accuracy. We conducted and confirmed each of these follow-up tests. Crucially, each of these tests was designed to answer the same basic question in a different way and was conducted for the express reason that we were not willing to conclude that sensory evidence was boosted on the basis of statistical significance of a single test, when there are additional well-powered follow-up tests available to confirm the effect. Indeed, this is one of the greatest advantages of our multi-level recording paradigm. Thus, the nature of these follow-up tests is such that their multiplicity greatly *lessens* the likelihood of type I errors, rather than increasing it. It should also be noted that our analyses of the behavioural data and of baseline beta-band activity constituted replications of previously well-established effects (e.g. Murphy et al., 2016, Nat. Comms.).

In summary, the multiple statistical tests in this paper were not each geared towards new discoveries of potentially fundamentally different effects, but rather constituted a conservative, comprehensive testing of one core hypothesis at multiple levels of processing (global signal enhancement). Where any effect diverged from this hypothesis (e.g. SSVEP underwent a differential, not common modulation by speed pressure), we followed up by conducting any available tests of the logical consequences of that effect in order to increase confidence in it. We thank the reviewer for raising this important point regarding risk of statistical errors and guiding us to clearly articulate the conservativeness of our approach in our new dedicated statistical analysis section (see next reply, and page 36 of the revised manuscript).

2) Although there are many regressions reported, it was not clear to me why these specifically were selected. For example, the specific choice for a random effects structure (only subject intercepts as far as I could tell) remained unmotivated, even though this is not considered the optimal way of including random effects. Similarly, the in- or exclusion of certain predictors seemed slightly adhoc (although to some extent consistent). Typically, model comparisons are reported to argue for or against including certain factors.

We thank the reviewer for raising this important concern regarding clarity and justification of statistical procedures. In the revised manuscript we have provided a thorough description and justification of our statistical methods in a new statistical analysis section in the Methods (page 36, relevant part pasted below). Further, we have improved our approach to including random effects in the way suggested by the reviewer; whereas originally we included only random intercepts across subjects, we now perform systematic model comparisons to determine for each test which random slopes factors are appropriate to include based on whether they contribute significantly to the model fit. Specifically, we employed an iterative approach, in which we initially computed a linear mixed-effects model without any random slopes and then included those random slopes that improved the model fit significantly when added. Following the suggestion of both Reviewers 1 and 2, we have also revised the statistics we report to affirm significant effects of each experimental factor throughout the paper, stating the chi-squared statistic for a model comparison of the model with vs. without each factor rather than the t-value as we had before (See also Reviewer 1, comment 5). We have pasted below the relevant part of the new Statistical analysis section of the methods, and we include all details of the iterative model selection process alongside the model equations in our new section *Supplementary statistical analysis details* in the Supplementary Materials. We thank the reviewer for guiding us toward this clearer, exhaustive description and conservative approach.

From page 36:

For all statistical tests involving multiple independent variables, we computed linear mixed-effects models on single-trial data if reaction time (RT) was a relevant independent variable, and otherwise used repeated-measures analyses of variance (ANOVA) on subject-averaged data. The use of linear mixed-effects models with the factor of RT was necessary to disentangle effects of RT arising from time-dependent influences within a trial (e.g. dynamic urgency) from other experimental factors that affected RT (e.g., Contrast, Speed/Accuracy Regime). To take a hypothetical example, if an identical collapsing bound were applied in the Speed and Accuracy Regime and it was through some mechanism other than bound adjustment that RTs were longer in the Accuracy Regime, the fact that the bound would have collapsed further with more elapsed time would mean that a basic comparison of an accumulator signal amplitude across Regimes would exhibit a difference, leading to an RT effect that masquerades as an amplitude (bound) effect. Controlling for such factors requires inclusion of the absolute value of RT as an independent variable rather than its rank within condition (e.g. quantile 1, 2, etc), necessitating the use of linear mixed effects models.

The conditional accuracy functions (Figure 1) revealed that response accuracy was particularly low for very early responses, peaked at approximately 600ms and thereafter

steadily decreased with increasing RT. This suggests that two distinct mechanisms may cause fast and late errors, respectively, and thus we allowed for non-monotonic relationships with RT in all linear mixed-effects models by including an RT^2 regression term in addition to RT itself. All linear mixed-effects models further included the fixed effect factors of Speed-vs-Accuracy emphasis (Regime), stimulus Contrast, and Left-vs-Right Trial Type to maintain consistency across measures. For the Steady-State Visually-Evoked Response (SSVEP) and the centroparietal positivity, the factor Left/Right distinguished trials based on whether the sensory evidence supported Left or Right choices. For motor level variables such as Mu/Beta amplitude and thumb EMG, Left/Right referred to whether subjects responded with their left or right hand.

In the linear mixed-effects models, we included random intercept terms to account for inter-subject variability in the dependent measures as appropriate for a repeated-measures design, and we included random slopes factors, which allow for differences in effect size across subjects, where their inclusion improved the model fit as assessed through systematic model selection. In contrast to a blanket policy of including all random slope terms by default (Barr et al., 2013), this data-driven approach has been demonstrated to retain the protection against type I errors while also avoiding increased type II errors (Bates et al., 2015; Matuschek et al., 2017). Specifically, we used an iterative process, in which we first computed a linear model without any random slopes and tested whether the model's fit could be significantly improved by including a random slope across each individual fixed effect factor, assessed using chi-squared tests. We provide an exhaustive account of this iterative model selection process in the *Supplementary statistical analysis details* section. Once the final model was established, we tested whether each fixed effects factor had a significant influence on the dependent variable by testing whether the fit of the model significantly worsened if it were excluded, providing the chi-square metrics stated throughout the results.

With these changes to the statistical approach, all of the previously stated effects remain significant, with the exception of the increase in Mu/Beta 'excursion' for the withheld response as a function of RT (Supplementary Figure 5b; now 5c). This test was carried out as a follow-up to the finding that the amplitude of the CPP at response decreases with increasing RT, in order to provide additional supporting evidence for a dynamic component of urgency at the motor level, which is already suggested by the fact that motor preparation signals for both the executed and withheld hand are launched in the same increasing direction upon evidence onset. Evidence for dynamic urgency has previously been found in monkey neurophysiology in the form of a significantly positive buildup slope component that was common to all motion stimulus conditions and manifest even in a condition with zero mean evidence (0% coherence; e.g. Churchland et al., 2008; Hanks et al., 2014). In the absence of a zero-evidence condition in our experiment, we instead explored for additional signs of dynamic urgency in the dynamics of motor preparation for the withheld response. Using similar reasoning to another recent human neurophysiology study claiming to observe dynamic urgency effects (Murphy et al., 2016), it could be expected that if Mu/Beta activity contralateral to the withheld response receives the same common component of urgency as the signal for the executed response, then it would grow to reach higher levels of preparation at the time of response on trials with longer RT. A weakness of this test, however, is

that it relies on the assumption that variations in amplitudes across trials with increasing RT can serve as a proxy for dynamics unfolding during a typical single trial, which is generally not the case. The relationship between the level reached by the “losing” motor alternative at response and RT would depend critically on which features of the decision process can randomly vary across trials and thereby influence RT, and also on what exactly drives the buildup of motor preparation for the the wrong alternative - e.g. urgency alone, negative sensory evidence, automatic co-activation with the opposite limb, or a mixture of these factors - none of which is fully known at present. For example, we found that slow RT trials were associated with relatively greater baseline motor preparation for the ultimately withheld response compared to the executed response (see Supplementary Figure 5, Supplementary Table 1k), and this would tend to reduce the possible excursion on those slower trials. As another example, if the urgency component common to both motor alternatives varied in steepness from trial to trial, then the steeper buildup of the losing motor alternative on shorter RT trials would counter the tendency for its level at response to increase over RT. In the revised manuscript we discuss the issues surrounding testing for dynamic urgency using Mu/Beta indices of preparation for the incorrect response, and discuss the range of other effects indicating its operation at the motor level, including the decline in the level reached by the CPP at response and building evidence-independent muscle activation, thus highlighting the value of the multi-tiered recording paradigm (see pages 21, 30 and 43).

3) Moving part of the details of the regression analyses and ANOVAs to the supplements is not a good choice, in my opinion. For me it became very difficult to follow the reasoning in the main text.

We thank the reviewer for this helpful observation. As stated above, we have added a designated section on statistical analyses to the main text (page 36), where we explain our reasoning for all choices of statistical test, including the use of linear mixed-effects models for analyses where RT is a critical factor. We have also reworked the results to ensure that the critical details needed to understand the test that was run (e.g., the type of statistical test run and the time window of measurement of the dependent variable) in each case is clear at the point where the test results are stated. As an example, in the results we had stated, regarding the differential SSVEP boost under Speed pressure, “This modulation was transient, emerging following evidence onset and lasting until just before the response (Supplementary Table 2d-e), suggesting that this differential boost was invoked specifically during decision formation.” As pointed out by Reviewers 1 (comment 7) and 3 (comment 3), this is an important result in the paper and it should be much clearer how exactly it was tested, without calling for a search through the supplementals. We therefore revised this section, which now reads, “This modulation [of differential SSVEP amplitude by speed pressure] was significant for all individual 400-ms windows centered from 350 to 550 ms (Figure 2b, top) and appeared to be limited to the period of decision formation, in that the Regime x Target Type interaction was significant just before the response (response-locked window centered at - 50ms, $F(1,15)=5.7$; $p=0.031$; Figure 2d; Supplementary Table 2e) but not just after (+ 50ms $F(1,15)=2.9$; $p=0.11$; Supplementary Table 2f).”

4) On page 8 the authors study the accuracy decline per unit time by doing a linear regression over all RTs *after* a participants' RT bin of maximum performance (or the mean RT in that bin). This analysis seems problematic, since the results depend on (a) The number of RT bins (which influences the estimation of the point of maximum performance and (b) the type of pooling of the data. It was not clear to me whether the pooling was done over Speed and Accuracy regimes, which seems to be problematic for detecting e.g. differences in the slope of the regression lines per speed/accuracy regime.

We agree with the reviewer that the motivation and details of this analysis were not sufficiently well described in the paper. The reviewer is correct in pointing out that the estimation of the point of maximum performance is to a degree dependent on the number of time bins chosen for this analysis. We chose to use six RT bins because it was the greatest number of RT-bins that provided a single, readily-detectable bin of peak performance. Nonetheless, to address the reviewer's concerns, we re-computed the point of maximum performance with different numbers of RT-bins ranging from 5 to 10. Although, as the reviewer predicted, the estimate of peak performance RT varied somewhat with respect to that obtained for 6 RT bins (difference of 7.5 +- 67.5 ms), the main result was always the same; that is, there was no significant effect of Speed/Accuracy Regime on the slope of declining accuracy as a function of RT. Note that as stated in our reply to Reviewer 1's comment 5, this was one of the two analyses that we revised to maintain consistency with our policy of using LME models only where effects of RT on the dependent measure need to be accounted for - in this case the dependent measure should be the rate of accuracy decline so we directly measured this by fitting a line to the CAF beyond the point of maximum performance and entering it into a repeated measures ANOVA. Critically, we repeated both the old LME test and the new ANOVA with cut-off RTs determined using different numbers of RT bins and the outcome was the same in all cases in that there was no main effect of Speed/Accuracy Regime on the CAF decline (5 RT bins: $F(1,15)=0.21$, $p=0.65$; 7 RT bins: $F(1,15)=0.12$, $p=0.73$; 8 RT bins: $F(1,15)=2.67$, $p=0.12$; 9 RT bins: $F(1,15)=0.15$, $p=0.70$; 10 RT bins: $F(1,15)=0.57$, $p=0.46$).

On the reviewer's second query regarding pooling, trials were pooled across speed and accuracy regimes and low and high contrast levels just for the purposes of finding the RT bin of peak performance. The test for differences in the slope across Speed/Accuracy Regimes used separate measures from the two Regimes and Contrast levels rather than pooling. To fully address the reviewers concern, we repeated the analysis using peak performance RTs obtained within each of the four individual conditions (which differed from the original estimate by 82.9+- 92.8ms) and again the results remained unchanged (main effect of Speed/Accuracy Regime: $F(1,15)=0.59$, $p=0.45$). We now explicitly lay out the details of this conditional accuracy function analysis in the new Statistical Analysis section, more clearly state the test used in the main text where we state the result, and include these additional checks to confirm robustness across other choices of RT bin number and pooling to compute peak performance RT on page 34-35.

5) On page 13, the authors report a repeated measures ANOVA to study how Speed/Accuracy regime predicts pupil size. The repeated measures in this analyses are the time points at which

the pupils size is determined. However, since these are not independent, estimates of the effect of the regimes are inflated (i.e., the standard errors are reduced), potentially leading to incorrect conclusions.

We thank the reviewer for this comment, which indicates to us that our phrasing of these effects in the main text, as well as the provision of statistical test results in the figure legend rather than the text, hampered the clarity of this result. We have now remedied this by moving the statistics back to the main text. The text now clearly states that the main effect of pupil size (increase under speed pressure) is the result of a t-test carried out on a single time window in the baseline period of -500-0 ms relative to evidence onset, which ensures that inflated estimates of Regime effects due to multiple time windows cannot occur. What was previously unclear was that our follow-up test of the time-dependence of this pupil effect of Regime was a separate 2-Way repeated-measures ANOVA with factors of Speed/Accuracy Regime and Time (30 50-ms time windows between 0 and 1500 ms after evidence onset). An interaction in this test indicates such time dependence.

Smaller points:

Page 3: “collapsing bound” is mentioned, but the uninformed reader might not know this concept.

Thank you for spotting this. We have added a more intuitive description of the concept as a **“decrease in the amount of accumulated evidence required to trigger a response over time (“collapsing bound” in computational models)”** to the section where we first mention collapsing bounds in the Introduction.

Page 8: It was not clear to me what the dependent variable was in the CAF regression analysis (also discussed above). Was this accuracy (correct/incorrect), or *proportion* correct per RT bin?

We now explicitly state that the dependent variable in the analysis of conditional accuracy functions was the proportion of correct trials within each response time bin per condition (see page 7). In response to comment #1 by Reviewer #1, we also elaborated on our explanation of conditional accuracy functions and their importance in studying time-dependent influences such as urgency.

Fig 3. “traces are baseline-corrected to the 500ms...” Does this mean this exact time point, or an average in the [500,0] ms interval?

An average in the 500-ms pre-cue interval. We thank the reviewer for pointing out that this was unclear. We now clarify that we baseline correct by subtracting the mean pupil size in the interval from -500ms to 0s with respect to Regime cue onset (see page 11).

Page 15: a p-value of $p=0.09$ cannot be considered a trend.

The reviewer is correct that an effect with $p=0.09$ (now $\chi^2(1)=3.11$, $p=0.078$ since we switched to reporting chi-squared statistics for model comparisons; see Reviewer 1 comment 5 above), should not be reported as a trend. Ideally, the time at which we measure CPP amplitude for this analysis should perfectly coincide with the time after which sensory evidence can no longer impact the decision, but this cannot be precisely ascertained. We therefore used the onset of the motor potential contralateral to the executed response as a reasonable *a priori* guide for the time at which to measure CPP amplitude and thereby chose a 60-ms time window centered on the onset of that motor potential (-100 ms). Given this timing uncertainty, and our subsequent observation that response-executing EMG bursts onset on average 35 ms earlier than this (see page 20), we tested several adjacent time windows to establish the reliability of the Regime and RT effects over a broader range of measurement windows, and found that the Regime effect reached significance for all windows centered on times between -170 and -130 ms and the RT effect for the range -130 to +110 ms (see Supplementary Figure 4). It was for this reason that we originally did not feel it appropriate to simply report a null result in the manuscript and referred instead to a 'trend'. This should all have been made far more explicit, and we have done so in our revision (see page 14). Rather than reporting a null result, we have opted to report the outcome of the statistical test for the *a priori* time window alongside the results from the adjacent windows so that the reader is fully informed.

Page 41: "accuracy deadline". It is not clear to me what the duration is that the evidence was on the screen. I think it 2.4 s, based on Fig 1a, but this could be specified.

We thank the reviewer for spotting this. We revised the text to clarify that the deadline in the "Accuracy deadline" condition indeed lies at 2.4s coinciding with the offset of evidence (see page 33).

Reviewer #3 (Remarks to the Author):

This manuscript describes a human electroencephalography (EEG) study which aims at characterizing how urgency affects perceptual decision-making at sensory, decision and motor processing stages. Their specific research question concerns whether urgency is applied "globally" at all processing stages, or whether distinct adjustments take place at different processing stages - a question grounded in previous, sometimes contradictory work on urgency. For this purpose, the authors use a well-controlled perceptual discrimination paradigm in which either the accuracy (low urgency) or the speed (high urgency) of performed decisions is emphasized through changing payoff strategies - as in previous work on this question. The authors show that urgency is reflected as a static increase in motor preparation (both cortical and muscular) even before the decision evidence is presented, but not reflected upstream in the neural correlates of decision evidence accumulation. Furthermore, the authors

report that the sensory processing of decision evidence is enhanced under high urgency to reduce the performance loss due to the shift in speed-accuracy tradeoff under temporal pressure.

I found the manuscript to be an interesting read, although difficult to follow at times, based on a well-designed paradigm and clever analyses of the recorded EEG data to decompose perceptual decision-making into sensory signals (steady-state visual evoked responses), decision evidence accumulation signals (the centro-parietal positivity component identified by the authors in earlier work) and motor preparation signals (band-limited mu/beta power overlying motor cortex, and electromyographic activity in thumbs used to report decisions). The pattern of findings reported in the manuscript is however complex and seemingly paradoxical at first glance, and thus bound to be fully grasped only by specialists. Therefore, at this stage, I am not entirely convinced that the obtained findings provide a sufficient advance in the existing knowledge about speed-accuracy trade-offs and urgency to justify publication in a general-audience journal - although I would be happy to revise my judgment

if the authors can increase the generality of the findings and their appeal to a non-specialist audience in a subsequent revision of the manuscript.

I have several comments, listed below, which I think the authors should consider for revising their manuscript. Some of these comments reflect issues that I have with reconciling the different, seemingly contradictory findings presented by the authors in the manuscript. Other reflect the incomplete description of the analyses applied to the data (e.g., the temporal windows of interest used for many of the statistical tests), which render their assessment very difficult - especially the use of mixed-effects models which appear unnecessary given the within-subjects design used by the authors.

We thank the reviewer for their careful review of the paper and for noting its strengths. As the reviewer notes, the results have a seemingly paradoxical aspect and the broad appeal of our findings strongly hinges on reconciling it in a very clear, accessible way. Thanks to the reviewer's detailed comments we have been able to pinpoint exactly how to do this. The apparent paradox is that under speed pressure the sensory evidence (SSVEP) is enhanced, yet the cumulative total of it (CPP) at the time of response is not increased. This is reconciled by the fact that enhanced evidence directly predicts a steeper buildup rate for the accumulator (CPP) - which is indeed what we found - but not necessarily a change to the peak level that it reaches at response. That peak level is instead determined by the action-triggering threshold crossing at the motor level. What we now make clearer, and will provide the "hook" for broad appeal, is that whereas the field has focused always on the deleterious effects of speed pressure on accuracy (i.e., the tradeoff), by recording at multiple sensorimotor processing levels we have revealed that accuracy-diminishing urgency effects at the motor level are applied in conjunction with accuracy-boosting enhancements at the sensory level. Although the motor-level adjustments ultimately play the dominant role in dictating behavioural outcomes, our analyses show that the accompanying sensory-level boost serves to limit the negative impact of urgency on choice accuracy. The representation of cumulative evidence (CPP) at the intermediate level exhibits the impact of both effects - though the sensory evidence enhancement steepens its buildup, the motor-level urgency curtails the level it can reach before a response is triggered. As described in more detail in the

individual replies below, we have clarified this message in the abstract, introduction and results section - particularly the schematic of Figure 5 which summarises how these seemingly paradoxical effects can coexist and account for the CPP observations. Also detailed below, we have also provided clearer, complete descriptions of the analyses and the motivations for statistical tests.

Note that below we have numbered the reviewer's comments for ease of reference.

Major comments:

1) * Introduction (p. 4): The authors draw a contrasted picture of the past literature on the effects of temporal pressure on the sensory processing of evidence during decision-making, suggesting that variations in the particular paradigms used to modulate urgency and in the neuroimaging modalities used to study it (animal electrophysiology, human fMRI, MEG/EEG) might explain some of the observed inconsistencies. Which such a line of reasoning can indeed explain the important differences observed across studies reported by the authors, it also suggests that the findings reported by the authors in the present study might be peculiar to the particular paradigm they used and the particular neuroimaging modality (EEG) they relied on. The authors should provide arguments as to why and how their findings (in particular the sensory enhancement of the differential evidence observed under urgency) are expected to generalize to other stimuli and manipulations of urgency.

There are indeed some apparent inconsistencies across previous studies using different techniques and tasks and we thank the reviewer for highlighting the need to clarify how our findings with this specific paradigm relate to these other studies, and whether, how and why our findings may be specific to the paradigm we employed. First, the behavioral task itself simply entails deciding between two alternative states of an elementary sensory feature, similar to any 2-alternative forced-choice task employed in the perceptual decision making field. There is no evidence we know of to suggest that when observers decide on the relative contrast of an object, they adapt their decision processes to speed pressure in a fundamentally different way to when they decide about motion, color, object identity (e.g. face versus car, face versus house), or any other dimension examined in decision research. Most importantly, the choice of contrast comparison as the task was key to being able to trace the timecourse of sensory evidence encoding in a way that has not been possible in any previous human research. Indeed, a crucial aspect of the study is the isolation of a sensory signal that specifically reflects the input to the decision process i.e. the 'evidence' - as distinct from just any sensory activity that may be evoked by the stimulus but is irrelevant to the decision (pages 7 and 31). Thus, the unprecedented nature of our finding is due to our paradigm design enabling an unprecedented view on sensory evidence dynamics, not due to the behavioral task itself being in any way peculiar.

Second, when one takes account of the well-known relative strengths and limitations of different techniques used, there is no study, at least that we know of, that clearly contradicts the results of our study. For example, our results are in agreement with the one electrophysiological study to date that has examined speed pressure effects on a similarly direct measure of evidence

encoding (Heitz and Schall, 2012), suggesting that quite different tasks (visual search versus 2-alternative sensory discrimination) invoke the same fundamental adjustment mechanisms for speed pressure. Our results represent an important novel contribution over this work by demonstrating that the effect generalizes from salience encoding neurons in monkey FEF to the representation of basic sensory attributes in human early visual cortex. The fMRI studies that have examined sensory activity changes in the speed-accuracy tradeoff have failed to find any change to the sensory representation of the evidence (Van Veen et al., 2008; Ho et al., 2012), but as we point out in the paper, the poor temporal resolution of fMRI means that it is not well suited to detecting transient modulations to sensory processing of the kind observed here. Further, there has yet to be an fMRI study that measures as direct a representation of sensory evidence as we have, through examining a feature that maps so readily onto activation amplitudes, such as contrast.

Finally, the vast majority of computational modelling studies that have manipulated speed pressure have reported no effect on the evidence-dependent drift rate parameter (e.g. Ratcliff and McKoon, 2008), which would indicate no effect at the level of sensory processing, and in the couple of instances where drift rate effects were found (Rae et al., 2014; Dambacher and Hübner, 2014) they were lower under speed pressure. However, behavioral data on their own are unlikely to provide sufficient constraints to be able to identify non-dominant effects that work against the dominant behavioral effect - that is, the lower drift rates in these cases simply reflect the lower overall accuracy under speed pressure. Further, a general limitation of behavioral modeling is that the validity of model selection outcomes critically depends on the accuracy with which the chosen models reflect the true dynamics of evidence encoding in the brain. To take a salient example, in the aforementioned study of monkeys performing visual search, fitting standard mathematical models (linear ballistic accumulator) to behavior indicated no drift rate differences, yet salience-encoding visual neurons of the frontal eye field exhibited increased and earlier target selectivity under speed pressure, consistent with a stronger evidence representation; and when a new mathematical model was constructed on the basis of neural observations, a drift rate effect fell out of the new behavioral model fit (Heitz and Schall, 2012). This highlights that abstract decision models, while providing parsimonious fits to behavioural data, often overlook important details of the underlying neural implementation. This underscores the fundamental importance of our demonstration that speed pressure can produce sensory level modulations, as it is based on experimental observations that reflect sensory evidence encoding more directly than any previous human study using other techniques. More generally, the multifaceted nature of the neural adjustments reported in our study resonate with growing concerns regarding the reliance on the assumption of 'selective influence' in the modelling literature whereby models that can account for particular cognitive processes by changes to a single parameter are considered superior to those that invoke multiple parameter changes (see Rae et al., 2014 for discussion). We are very grateful to the reviewer for highlighting these important points which should have been more clearly discussed in the original manuscript, and are now incorporated in the revised text on pages 4, 7, 23 and 24.

2) * Abstract (p. 2) and Introduction (p. 7): Since the representation of cumulative evidence is found downstream from the sensory processing of the evidence samples, it is hard to understand from the abstract how to reconcile an enhanced sensory representation of the differential evidence (upstream) with a weaker representation of the cumulative evidence (downstream). A standard prediction would be that because of the increased representation of the differential evidence under temporal pressure, the ramping up of the cumulative evidence should be increased and thus reach the same level at motor response onset as in the non-speeded condition. The authors might want to re-write these sections in such a way that does not trigger such apparent inconsistency - which is later considered and discussed by the authors.

This is extremely valuable feedback; as we acknowledged above, providing a clear picture of how these different effects fit together in the data is critical to the impact of the findings reaching a broad audience. The critical detail that was originally missing is that in this evidence accumulation framework, enhanced sensory evidence strength should directly translate to a steeper buildup rate of the accumulator signal (which it does), whereas it is the motor-level urgency, which elevates motor signals towards their fixed thresholds, that determines the level of cumulative evidence that can be reached before an action is triggered. We have revised the short summary of results we give in the Introduction to make this explicitly clear (**“The knock-on impact of this sensory enhancement was exhibited in a steepening of evidence accumulation reflected in the CPP. Meanwhile, evidence-independent motor-level urgency, which was elevated under speed pressure but appeared to grow at a similar rate under the Speed and Accuracy regimes, limited the amplitude that the CPP reached by the time a response was triggered.”**, page 6), and have improved the clarity of our Figure 5 schematic (panel c) which illustrates how the two separate knock-on effects of enhanced sensory evidence and motor-level urgency can result in a CPP buildup that is steepened yet still reaches lower levels at the point of decision termination under speed pressure.

3) * Results (p. 10): The authors often do not describe precisely the time window at which differential evidence is "boosted under speed pressure". This is a general comment about the reporting of findings throughout the Results section, where important details are often missing. Also, the authors make a crucial claim about the transient nature of this effect, restricted to the evidence accumulation period, but mention no statistics to support this claim, no illustration of this effect and refer merely to a supplementary table. Because this sensory enhancement under speed pressure is one of the most surprising (and novel) effects reported in the manuscript, the authors should in my opinion provide all the necessary information to describe this effect.

We thank the reviewer for pointing out this issue. We have thoroughly revised the Results to make sure that important details such as the time window of measurement are mentioned explicitly for each test.

With respect to the transience of the sensory enhancement, we have now clarified the text the reviewer mentions to make it clear that the tests referred to originally in the supplementary tables simply test for the Regime effect on the differential evidence in the timepoint just before and in the timepoint just after the time of response, a clarification also called for by reviewer 1 (comment

7). The reviewer is correct that for such an important effect we should display the data more exhaustively, and we have thus revised the figure to include the response-locked waveforms as well. The new figure is pasted below. Note that the SSVEP amplitude is measured in a 400-ms window and thus ramps to reach steady-state at least 200 ms after evidence onsets (Figure 2b). Differences in RT between the experimental conditions then lead to the apparent differences in the rate of rise in the response-locked traces (Figure 2d).

Note also that the revised figure shows a revised timecourse of F-values for the Regime x Left/Right interaction defining the differential sensory evidence boost. Whereas we had previously conducted these time-resolved tests on averages of 4 consecutive 400-ms windows, we now conduct the tests on each individual 400 ms FFT window to maximise temporal resolution. This has no bearing on the results qualitatively.

4) * Results (p. 11): I do not understand the motivation underlying the use of a mixed-effects model throughout the Results section. Indeed, the full experimental design appears to be using only within-subjects factors, which affords the use of classical repeated-measures ANOVAs and does not require the use of mixed-effects models. The authors never justify the use of a mixed-effects models, neither in the Results nor in the Methods sections, something which is annoying given that such a statistical model does not seem fully warranted given their experimental design (at least as far as I can tell). Another concern is whether the reported statistics from the mixed-effects model are random-effects statistics (which take the variability across tested subjects as measure of dispersion) or fixed-effects statistics (which take the variability within tested subjects as measure of dispersion). Again, this important point is not described anywhere in the Methods or even Supplementary Material sections. I therefore cannot assess the validity of the statistical analyses relying on mixed-effects models, and remain doubtful regarding the strength of the reported effects based on such models.

We appreciate the reviewer raising this critical issue regarding an incomplete description and justification of the statistical approach we took, which was also raised by both Reviewer 1 (comment 5) and Reviewer 2 (comment 2). As indicated in our replies above, we have thoroughly revised the way in which we present our statistical methods and the reasoning behind their use in a new dedicated statistical analysis section of the Methods and have also improved the reporting of statistics throughout the results. As described below in that new section, we used

LME models in all cases where RT was a relevant independent variable, or where its relationship with the dependent variable represents a potential confounding factor for assessing other factors. We also state our revised approach of including random slopes factors where their inclusion significantly contributed to model fit, and our revised reporting of chi-squared statistics throughout the results for assessing the significant contribution of each fixed-effects factor.

From page 36:

For all statistical tests involving multiple independent variables, we computed linear mixed-effects models on single-trial data if reaction time (RT) was a relevant independent variable, and otherwise used repeated-measures analyses of variance (ANOVA) on subject-averaged data. The use of linear mixed-effects models with the factor of RT was necessary to disentangle effects of RT arising from time-dependent influences within a trial (e.g. dynamic urgency) from other experimental factors that affected RT (e.g., Contrast, Speed/Accuracy Regime). To take a hypothetical example, if an identical collapsing bound were applied in the Speed and Accuracy Regime and it was through some mechanism other than bound adjustment that RTs were longer in the Accuracy Regime, the fact that the bound would have collapsed further with more elapsed time would mean that a basic comparison of an accumulator signal amplitude across Regimes would exhibit a difference, leading to an RT effect that masquerades as an amplitude (bound) effect. Controlling for such factors requires inclusion of the absolute value of RT as an independent variable rather than its rank within condition (e.g. quantile 1, 2, etc), necessitating the use of linear mixed effects models.

The conditional accuracy functions (Figure 1) revealed that response accuracy was particularly low for very early responses, peaked at approximately 600ms and thereafter steadily decreased with increasing RT. This suggests that two distinct mechanisms may cause fast and late errors, respectively, and thus we allowed for non-monotonic relationships with RT in all linear mixed-effects models by including an RT^2 regression term in addition to RT itself. All linear mixed-effects models further included the fixed effect factors of Speed-vs-Accuracy emphasis (Regime), stimulus Contrast, and Left-vs-Right Trial Type to maintain consistency across measures. For both RT and all neurophysiological measures the z-score was computed before being entered into the model. For measures of motor preparation this z-score was computed separately for the left and right hemisphere. For the Steady-State Visually-Evoked Potential (SSVEP) and the centroparietal positivity, the factor Left/Right distinguished trials based on whether the sensory evidence supported Left or Right choices. For motor level variables such as Mu/Beta amplitude and thumb EMG, Left/Right referred to whether subjects responded with their left or right hand.

In the linear mixed-effects models, we included random intercept terms to account for inter-subject variability in the dependent measures as appropriate for a repeated-measures design, and we included random slopes factors, which allow for differences in effect size across subjects, where their inclusion improved the model fit as assessed through systematic model selection. In contrast to a blanket policy of including all random slope terms by default (Barr et al., 2013), this data-driven approach has been demonstrated to retain the protection against type I errors while also avoiding increased type II errors

(Bates et al., 2015; Matuschek et al., 2017). Specifically, we used an iterative process, in which we first computed a linear model without any random slopes and tested whether the model's fit could be significantly improved by including a random slope across each individual fixed effect factor, assessed using chi-squared tests. We provide an exhaustive account of the step-by-step results of this iterative model selection process in the *Supplementary statistical analysis details* section. Once the final model was established, we tested whether each fixed effects factor had a significant influence on the dependent variable by testing whether the fit of the model significantly worsened if it were excluded, providing the chi-square metrics stated throughout the results. For all tests we used an alpha level of 0.05.

5) * Results (p. 12): As indicated earlier, the authors do not report the latency at which they observed pupil effects, and described the findings too succinctly to assess the strength of the reported effect.

Following the reviewer's feedback and a related comment of reviewer 2 (comment 5), we have completely clarified this section by specifying that we first conducted a t-test on pupil size measured in the baseline period of -500 to 0ms relative to evidence onset and then a 2-Way repeated-measures ANOVA to test whether the Regime effect increases with time. The analysis of the relationship between pupil size and SSVEP amplitude was based on the mean pupil size between 0ms and 1500ms after stimulus-onset (see page 45-46).

6) * Results (p. 17, "The decreased peri-response amplitude [...] directly at the CPP level." I am unconvinced that narrowing the bounds is necessarily "dynamic" per se. In my opinion, it can be a static effect that is dissociable from the baseline increase in motor activity. Also, this effect seems to be present for both speed and accuracy conditions - when I was expecting to see it only in the speed (urgency) condition. Can the authors elaborate on this theoretical point, and explain why their results truly support a dynamic view rather than a static change in decision bound?

The reviewer is correct that in general, significant variation of amplitude across trials sorted by RT does not in itself indicate that there is necessarily a "dynamic" change occurring within each individual trial. However, if the variation of CPP amplitude with RT were due to variation of a static bound across trials, then the direction of the effect would be opposite to what was observed - trials with longer RT would be associated with relatively wide static bounds and shorter RTs with relatively narrow static bounds. The significant, random trial-to-trial variability in starting level of the motor preparation signals, in fact, effectively amounts to such variability in a static component of bound height, and indeed produces lower accuracy and lower CPP amplitudes for the trials with the very fastest RT in the Speed Regime, compared to trials with slightly longer RTs (see Fig 1c and Fig 4h). However, apart from those very earliest RT bins in the Speed Regime, both accuracy and CPP amplitude at the time of decision commitment decline rather than increase over RT, which cannot be generated by trial-to-trial variability in either the baseline motor

preparation or a static bound. We now make these arguments explicit in the revised manuscript (page 17).

The reviewer also mentions the fact that the decrease in CPP amplitude (and also accuracy) over RT is observed in the Accuracy as well as the Speed Regime. This is indeed true and is a potential source of confusion when digesting the results of the study. As we explicitly lay out in our schematic of fig 5 summarizing our findings (now revised to enhance clarity), our results indicate that the static (time-independent, i.e., constant offset) component of urgency is adjusted between the Speed and Accuracy regimes, but the dynamic component of urgency, defined by the slope with which urgency increases over time, appears to be similarly steep in both regimes. The likely explanation for this is that the speed-accuracy regime was cued on a trial-to-trial basis and even the Accuracy Regime involved some speed pressure due to the ultimate 2.4-second deadline and the decline in points awarded as a function of RT in one of the Accuracy emphasis manipulations. This is interesting in itself, because it may indicate that while participants could raise their baseline motor preparation and arousal in response to the speed emphasis cues, they may not have been capable of adjusting the rate of urgency growth over time from one trial to the next. We have revised the summaries of results both in the Introduction and in Figure 5 to make sure this distinction between static and dynamic urgency adjustments is explicitly clear.

7) * Figures: Wherever possible, the authors should display error bars (e.g., s.e.m.) rather than only the mean across tested subjects. I can understand that individual subject data does not need to be plotted to assess the significance of an effect, but a measure of between-subject variability such as the s.e.m. is desperately needed to be shown for readers to assess (at least qualitatively) the strength of the reported findings.

We agree that error bars are very important. In addition to the error bars already displayed in Figures 1, 4 and 6, we now include error bars for measures of the mean stimulus-locked SSVEP amplitude and pupil diameter at baseline in Figures 2 and 3, respectively.

8) * Results (p. 20, "responses are initiated while the evidence accumulation is ongoing"): The authors should avoid making too clear/strong reverse inference here. They should reformulate the cited statement as "[...] while the neural correlates of evidence accumulation are still present", especially because the statement is found in the Results section, not in the Discussion section where such reverse inference might be more appropriate.

We agree. We have revised the statement to read "...suggesting that even for simple button presses, responses are initiated while the neural correlates of evidence accumulation are still evolving"

Minor comments:

* Introduction (pp. 5-6, "Resolving all of these questions [...] into motor preparation."): I found this paragraph which describes the different EEG measures of sensory processing, evidence accumulation and response preparation to be too long. The authors should in my opinion shorten this paragraph.

We agree that this paragraph was laborious as written and have made revisions to shorten it and break it up for better readability while retaining the critical information required to make our case for using these signals to study the various levels of processing involved in decision making.

* Results (p. 7): I am unclear as to how a "blocked" urgency design across trials could not have afforded to examine the effects of urgency at each neural processing level - especially because the identification of these different neural processing levels is done based on a "a priori" definition of particular neural signals that seems entirely unrelated to the "blocked" or "interleaved" nature of the experimental design. Can the authors either reformulate or remove this statement?

We assume that the reviewer is here referring to our statement "Randomized trial-by-trial cueing was used so that the short-term establishment of pre-decision states in preparation for speed pressure could be examined at each neural processing level." The reviewer is correct in asserting that the same neural signals can be measured in a "blocked" design as much as this interleaved design. However, whereas spectral measures such as Mu/Beta amplitude can be measured without referencing to a baseline, broadband ERPs need to be referenced to a baseline interval in the recent past to minimise the impact of spurious slow electrode drift. Baseline-correcting our CPP waveforms with respect to the signal amplitude just before the Speed/Accuracy Regime-cue (when the regime is not yet known) allowed us to test for potential strategic pre-evidence starting point adjustments of the CPP, whose absence was critical in concluding that urgency is applied at the motor preparation level but not upstream evidence accumulation. In a "blocked" design, the Regime would be known to the subject throughout the block and strategic adjustments before individual trials would not have been necessary. We have thus reformulated this sentence to read, "Randomized trial-by-trial cueing was used so that the short-term establishment of pre-decision states in preparation for speed pressure could be examined with respect to a recent, pre-cue baseline."

* Figure 1: The auditory cue should be shown as one line on panel (a). It is a very important feature of the paradigm which is apparently missing from the figure.

The reviewer is correct in pointing out the importance of an accurate depiction of the experimental design and we have revised Figure 1 to include the timing of the auditory cue.

* Discussion (p. 27): It is unclear what the authors mean by "collapsing decision bound": do the authors mean a collapsing bound over time within a trial (which I don't think there is any evidence for in the reported findings, but which is what is meant by collapsing bound in computational

modeling papers), or simply an overall lower decision bound which does not shrink over time within each trial?

We thank the reviewer for motivating us to clarify this important aspect of the interpretation of our data. We do indeed mean a collapsing bound over time within a trial, and as we state in our reply to comment 6 above. We believe the source of confusion is that the rate of collapse does not seem to differ between the Speed and Accuracy Regimes. We have now taken greater care in our introduction and results to make the distinction that our finding of a growing component of urgency (in addition to the static component) is not one of the adjustments made between Regimes but is a feature of both regimes. Several aspects of our data indicate that the effective decision bound is collapsing within a trial for both Regimes. Despite the motor level signals reaching a fixed amplitude at the time of response regardless of Regime or RT, both the CPP amplitude just before the response time and choice accuracy declined as a function of RT, indicating that the contribution of accumulated evidence to the decision decreased over response time. Meanwhile muscle activation for the withheld response was increased for longer response times. This is all consistent with the collapsing bound being implemented via a growing urgency component applied at the motor level (see also e.g. Churchland et al., 2008 and Hanks et al., 2014), and as we state above in our reply to comment 6, is wholly inconsistent with alternative accounts involving variability in static urgency components or time-independent bounds.

REVIEWERS' COMMENTS:

Reviewer #1 (Remarks to the Author):

The paper has improved a lot! I only have a few minor comments remaining.

1) I really appreciate that the authors now use Bayes Factors to indicate equivalence of results. However, in a few cases these Bayes Factors are still missing:

- p.9-10 (SSVEP amplitudes showed no main effect of phase-reversal frequencies)

-p. 35 (no main effect of Speed/Accuracy regime on CAF decline).

2) Also, it would be good to indicate in the methods (1) what software you used for the computation of Bayes Factors (and actually, for statistics in general), and (2) a brief discussion of how to interpret these bayes Factors for readers not so familiar with these statistics.

3) p.7 "The latter is consistent with the presence of a dynamically growing urgency component." Could you explain how this follows from accuracy decreasing monotonically at a similar rate for both regimes.

4) In Figure 2 I do not see dashed vertical lines indicating the stimulus or response

5) p.30 "implying that this finding does not necessarily generalize to all actions." I think this conclusion is too strong, because there are many other differences between human and primate studies (for one thing: primates are severely water-deprived)

The abbreviation CAF is never explained

Reviewer #2 (Remarks to the Author):

This is a revision version of a paper that I reviewed some time ago. The authors have done a great job in improving readability of the manuscript, which will likely make the paper much more accessible for a broad audience. I also appreciate the improved clarity in the statistical analyses, and in particular the discussion of the methodological considerations in the Methods section (e.g., why linear mixed effects are used).

I still worry about the large number of statistical analyses in this paper in relation to the chance of a false positive (indeed, in my previous review I mistakenly said "Type II" rather than "Type I" error). The authors argue that the repeated finding of significant effects actually decreases the Type I error rate, since the follow-up tests consistently replicate the findings of the primary tests. I believe this argument only holds if the primary and follow-up tests are based on independent data sets, which is not the case here. Imagine the scenario where the participants in this study are all outliers of the population in the sense that they show the reported effects, but nobody else does. Given the small sample size of 16 participant it is not completely unlikely that a slightly less extreme version of this scenario is true. In that case, the significant effects of the primary tests will propagate to the follow-up tests, but will not generalize to the population. This example refutes the argument that the follow-up tests decrease the Type I error rate. Depending on the strength of the dependence between the various tests, the probability of Type I error may increase, stay about equal, or decrease (for independent samples/tests).

One way out of this problem is to acknowledge the exploratory nature of the current study,

and to plan a confirmatory (possibly pre-registered) follow-up experiment. For the current manuscript, I think the authors should either argue/show that the evidence is so strong that it survives a correction for multiple comparisons, or discuss the limits that the sample size places on the conclusions. The fact that this studies replicates earlier findings from the literature is of course reassuring and could be emphasized (as the authors already do in this revision).

There remain some small mistakes in the manuscript:

Page 7: I think the authors report the statistics based on 6 RT bins rather than 8. At least, the reported F and p-values are not consistent with those reported in the rebuttal.

Caption of Figure 2: Fig 2b mentions a Dashed vertical line, but this is not in the figure. Perhaps the problem is in the lettering of the panels. The figure has panels a-c, but the caption mentions a-d.

Supplementary Figure 4: "the significant effect of reaction time on CPP amplitude at response is stable over different time windows of measurement". I could use some more explanation about this statement. The orange line in the upper panel fluctuates quite a lot. I assume the authors mean that this RT effect is stable for a certain range of time windows?

REVIEWERS' COMMENTS:

Reviewer #1 (Remarks to the Author):

The paper has improved a lot! I only have a few minor comments remaining.

1) I really appreciate that the authors now use Bayes Factors to indicate equivalence of results. However, in a few cases these Bayes Factors are still missing:

- p.9-10 (SSVEP amplitudes showed no main effect of phase-reversal frequencies)

-p. 35 (no main effect of Speed/Accuracy regime on CAF decline).

We thank the reviewer for pointing this out and now report the results of four Bayes Factor analyses evaluating the relationship between speed pressure and the amplitude of single SSVEP frequencies (page 8).

“SSVEP amplitudes for individual phase-reversal frequencies showed no main effect of speed pressure either before evidence onset (t-tests on window -400 to 0 ms; 20Hz: $t(15)=0.4$, $p=0.68$; $BF_{01}=3.610$; 25Hz: $t(15)=0.5$, $p=0.61$; $BF_{01}=3.467$) or in the decision-formation time range over which the above sensory boost effect (Regime x Target Type interaction) was significant (3-Way repeated-measures ANOVA, 350-550 ms; 20Hz: $F(1,15)=0.6$, $p=0.46$; $BF_{01}=4.969$; 25Hz: $F(1,15)=0.3$, $p=0.61$; $BF_{01}=5.493$; Supplementary Table 4g-h, See Supplementary Fig. 2 for time-resolved analysis).”

The Bayes Factor of the decline in the conditional accuracy function is being reported on page 7.

“The rate of accuracy decline over RT **did not** differ **significantly** as a function of Regime (slope of a line fit to response accuracy over 8 RT bins lying beyond each subject's point of maximum response accuracy, ANOVA, $F(1,15)=1.7$, $p=0.22$, Supplementary Table 4c), although the evidence for equivalence was not strong ($BF_{01} = 2.197$).”

2) Also, it would be good to indicate in the methods (1) what software you used for the computation of Bayes Factors (and actually, for statistics in general), and (2) a brief discussion of how to interpret these bayes Factors for readers not so familiar with these statistics.

We have added a paragraph to the Methods in which we state that the Bayes Factors were computed using JASP and give a brief introduction into the nature of this statistical test. All other statistical tests were computed using Matlab functions available in the statistics toolbox.

3) p.7 "The latter is consistent with the presence of a dynamically growing urgency component." Could you explain how this follows from accuracy decreasing monotonically at a similar rate for both regimes.

We thank the reviewer for pointing out the potential confusion in this sentence. It is the accuracy decline itself, not the fact that its rate does not differ between regimes, that indicates the operation of a growing urgency component, and we have now amended that sentence to clarify this.

To be clear, in this paper, we are defining “urgency” as evidence-independent, additive contributions to the decision process. The main characteristics of such signals is that they affect both response alternatives equally and that they do in no way interact with the physical evidence or its neural representation. In this definition, we include contributions that do not vary across Speed and Accuracy emphasis. The dynamic urgency component we are reporting raises motor preparation towards both response alternatives towards action triggering bounds as each trial progresses. Such an evidence-independent increase in motor preparation over the course of a trial implies that less and less accumulated evidence is required to trigger a response. A lower amount of accumulated evidence, in turn goes hand-in-hand with a decrease in response accuracy. We therefore interpret the observed decrease in response accuracy for longer reaction times as an indicator for the presence of a dynamic urgency contribution in both the Speed and the Accuracy regime. This interpretation is later confirmed by the decrease in CPP amplitude at response over reaction time. We are confident that this is now fully clear in the paper.

4) In Figure 2 I do not see dashed vertical lines indicating the stimulus or response

We thank both reviewers for pointing this out. We have now updated the figure so that the stimulus-locked traces include dashed lines indicate mean response times per condition.

5) p.30 "implying that this finding does not necessarily generalize to all actions." I think this conclusion is too strong, because there are many other differences between human and primate studies (for one thing: primates are severely water-deprived)

We agree with the reviewer that there are differences between different tasks that go beyond the actions used to report choices. We have now adjusted the text to include the possibility of such differences.

"In contrast to our findings of shortened and intensified muscle activation, work in primates suggests no differences in saccade velocity as a function of RT or speed pressure²², implying that this finding does not necessarily generalize to all actions, although species differences may also be relevant here. "

The abbreviation CAF is never explained

We have removed the abbreviation from the text and are now writing-out "conditional accuracy function".

Reviewer #2 (Remarks to the Author):

This is a revision version of a paper that I reviewed some time ago. The authors have done a great job in improving readability of the manuscript, which will likely make the paper much more accessible for a broad audience. I also appreciate the improved clarity in the statistical analyses, and in particular the discussion of the methodological considerations in the Methods section (e.g., why linear mixed effects are used).

I still worry about the large number of statistical analyses in this paper in relation to the chance of a false positive (indeed, in my previous review I mistakenly said "Type II" rather than "Type I" error). The authors argue that the repeated finding of significant effects actually decreases the Type I error rate, since the follow-up tests consistently replicate the findings of the primary tests. I believe this argument only holds if the primary and follow-up tests are based on independent data sets, which is not the case here. Imagine the scenario where the participants in this study are all outliers of the population in the sense that they show the reported effects, but nobody else does. Given the small sample size of 16 participant it is not completely unlikely that a slightly less extreme version of this scenario is true. In that case, the significant effects of the primary tests will propagate to the follow-up tests, but will not generalize to the population. This example refutes the argument that the follow-up tests decrease the Type I error rate. Depending on the strength of the dependence between the various tests, the probability of Type I error may increase, stay about equal, or decrease (for independent samples/tests).

One way out of this problem is to acknowledge the exploratory nature of the current study, and to plan a confirmatory (possibly pre-registered) follow-up experiment. For the current manuscript, I think the authors should either argue/show that the evidence is so strong that it survives a correction for multiple comparisons, or discuss the limits that the sample size places on the conclusions. The fact that this studies replicates earlier findings from the literature is of course reassuring and could be emphasized (as the authors already do in this revision).

The reviewer is correct that although the internal consistency of the results in the current paper shows that the effects are reliable in this sample, we cannot guarantee that the exact same results would be obtained in another independent sample of participants - this is probable, but not definite, and just like any other study it will be important to replicate the results. We have highlighted this in our discussion, in particular pointing to extended conditions under which it will be useful to confirm the operation of these same mechanisms, e.g. for other feature discriminations such as motion, and in conditions with zero evidence (see revised discussion). In fact, we have ongoing studies addressing these extensions and exhibiting results in line with those in the present paper, which we are preparing for separate publication.

There remain some small mistakes in the manuscript:

Page 7: I think the authors report the statistics based on 6 RT bins rather than 8. At least, the reported F and p-values are not consistent with those reported in the rebuttal.

We apologize if this was still unclear. In the paper's main text (page 7), we report the results of one 3-Way repeated-measures ANOVA which tests the effect of Contrast, Target Type and Speed/Accuracy Regime on the decrease in response accuracy over RT. For this analysis, we first divide all trials per subject into 6

RT bins and determine the cut-off RT as the mean RT in the bin with the greatest response accuracy. We then go on and divide trials within each condition with RTs slower than the cut-off RT into 8 RT bins, fit lines to the resulting performance-over-RT functions and test for a significant influence of Speed/Accuracy Regime on the slope of this line using the 3-Way ANOVA.

As the reviewer pointed out in a previous comment, it is conceivable that the presence or absence of such an effect would depend on the cut-off RT. To demonstrate that the null result stands even if cut-off RTs are (1) measured within each experimental condition rather than in RT data pooled across conditions within a subject, and (2) determined based on a different number of reaction time bins (5 through 10 RT bins), we repeated the analysis for all above cases. We report the results of these control conditions in the rebuttal as well as the Methods section of the paper (page 25).

Caption of Figure 2: Fig 2b mentions a Dashed vertical line, but this is not in the figure. Perhaps the problem is in the lettering of the panels. The figure has panels a-c, but the caption mentions a-d. We have now updated Figure 2 to show panels a through d and the vertical dashed lines.

Supplementary Figure 4: “the significant effect of reaction time on CPP amplitude at response is stable over different time windows of measurement”. I could use some more explanation about this statement. The orange line in the upper panel fluctuates quite a lot. I assume the authors mean that this RT effect is stable for a certain range of time windows?

Yes, this is what we meant and we have now clarified this by specifying this range of time windows. The reviewer is correct in their observation that both the effect of RT and speed pressure on the CPP amplitude at response are not significant throughout the trial. Instead, the amplitude of the CPP is greater for trials with slow reaction times at the beginning of the response-locked epoch, as some time points precede evidence-onset on trials with faster responses. Around the time of the response, RT then has a negative effect on CPP amplitude in line with an increasing influence of motor urgency. Speed pressure only decreases the response-aligned CPP around the time of decision commitment. In line with the suggestion that there is more post-decision signal processing under speed pressure, the CPP peaks closer to the button click under speed pressure. This gives the CPP amplitudes which are still significantly lower around the time of decision commitment time to “overtake” the CPP measured under Accuracy emphasis, so that the effect of speed pressure on CPP amplitude is non-significant around the time of the button click, before the effect reverses and CPPs measured under speed pressure maintained significantly greater amplitudes than those measured under Accuracy emphasis.